# PANORAMA: FAST-TRACK NEAREST NEIGHBORS

## ABSTRACT

Approximate Nearest-Neighbor Search (ANNS) efficiently finds data items whose embeddings are close to that of a given query in a high-dimensional space, aiming to balance accuracy with speed. Used in recommendation systems, image and video retrieval, natural language processing, and retrieval-augmented generation (RAG), ANNS algorithms such as IVFPQ, HNSW graphs, Annoy, and MRPT utilize graph, tree, clustering, and quantization techniques to navigate large vector spaces. Despite this progress, ANNS systems spend up to 99% of query time to compute distances in their final *refinement phase*. In this paper, we present PANORAMA, a machine learning-driven approach that tackles the ANNS verification bottleneck through data-adaptive learned orthogonal transforms that facilitate the accretive refinement of distance bounds. Such transforms compact over 90% of signal energy into the first half of dimensions, enabling early candidate pruning with partial distance computations. We integrate PANORAMA into SotA ANNS methods, namely IVFPQ/Flat, HNSW, MRPT, and Annoy, without index modification, using level-major memory layouts, SIMD-vectorized partial distance computations, and cache-aware access patterns. Experiments across diverse datasets—from image-based CIFAR-10 and GIST to modern embedding spaces including OpenAI's Ada 2 and Large 3—demonstrate that PANORAMA affords a 2-30x end-to-end speedup with no recall loss.

## 1 INTRODUCTION AND RELATED WORK

The proliferation of large-scale neural embeddings has transformed machine learning applications, from computer vision and recommendation systems (Lowe, 2004; Koren et al., 2009) to bioinformatics (Altschul et al., 1990) and modern retrieval-augmented generation (RAG) systems (Lewis et al., 2020; Gao et al., 2023). As embedding models evolve from hundreds to thousands of dimensions—exemplified by OpenAI's `text-embedding-3-large` (Neelakantan et al., 2022)—the demand for efficient and scalable real-time Approximate Nearest-Neighbor Search (ANNS) intensifies.

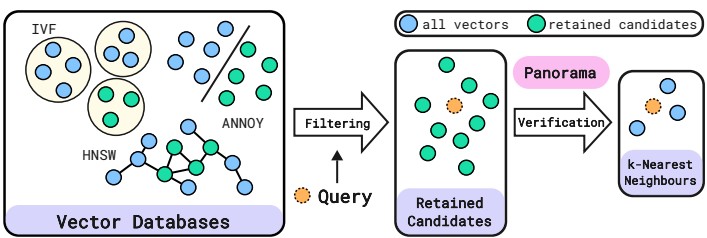

Figure 1: Common ANNS operations on vector databases.

Current ANNS methods fall into four major categories: *graph-based*, *clustering-based*, *tree-based*, and *hash-based*. Graph-based methods, such as HNSW (Malkov & Yashunin, 2020) and DiskANN (Subramanya et al., 2019), build a navigable connectivity structure that supports logarithmic search. Clustering and quantization-based methods, e.g., IVFPQ (Jégou et al., 2011; 2008) and ScaNN (Guo et al., 2020), partition the space into regions and compress representations within them. Tree-based methods, including kd-trees (Bentley, 1975) and FLANN (Muja & Lowe, 2014), recursively divide the space but degrade in high dimensions due to the *curse of dimensionality*. Finally, hash-based methods, such as LSH (Indyk & Motwani, 1998; Andoni & Indyk, 2006) and multi-probe LSH (Lv et al., 2007), map points into buckets so that similar points are likely to collide. Despite this diversity, all such methods operate in two phases (Babenko &

Lempitsky, 2016): *filtering* and *refinement* (or verification). Figure 1 depicts this pipeline. Filtering reduces the set of candidate nearest neighbors to those qualifying a set of criteria and *refinement* operates on these candidates to compute the query answer set. Prior work has overwhelmingly targeted the filtering phase, assuming that refinement is fast and inconsequential. This assumption held reasonably well in the pre–deep learning era, when embeddings were relatively low-dimensional. However, neural embeddings have fundamentally altered the landscape, shifting workloads toward much higher dimensionality and engendering a striking result shown in Figure 2: *refinement now accounts for a dominant 75–99% share of query latency, and generally grows with dimensionality.* Some works sought to alleviate this bottleneck by probabilistically estimating distances through partial random (Gao & Long, 2023) and PCA projections (Yang et al.,

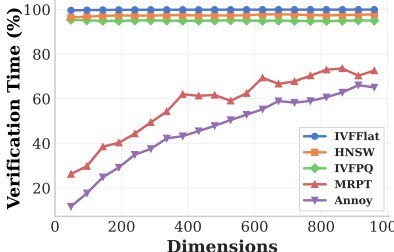

Figure 2: Time share for refinement.

2025) and refining them on demand. However, such probabilistic estimation methods forgo exact distances and, when using random sampling, preclude any memory-locality benefits. This predicament calls for innovation towards efficient and exact refinement in ANNS for neural embeddings. In this paper, we address this gap with the following contributions.

- **Cumulative distance computation.** We introduce PANORAMA, an accretive ANNS refinement framework that complements existing ANNS schemes (graph-based, tree-based, clustering, and hashing) to render them effective on modern workloads. PANORAMA incrementally accumulates $L_2$ distance terms over an *orthogonal transform* and refines lower/upper bounds on the fly, promptly pruning candidates whose lower distance bound exceeds the running threshold.

- **Learned orthogonal transforms.** We introduce a data-adaptive *Cayley transform* on the Stiefel manifold that concentrates energy in leading dimensions, enabling tight *Cauchy–Schwarz distance bounds* for early pruning. Unlike closed-form transforms, this learned transform adapts to arbitrary vector spaces, ranging from classical descriptors like SIFT to modern neural embeddings.

- **Algorithm–systems co-design.** We carefully co-design system aspects with specialized variants for contiguous and non-contiguous memory layouts, leveraging *SIMD vectorization*, cache-aware layouts, and batching, and also provide theoretical guarantees alongside practical performance.

- **Integrability.** We fold PANORAMA into five key ANNS indexes (IVFPQ, IVFFlat, HNSW, MRPT, Annoy) to gain speedups without loss of recall and showcase its efficaciousness through experimentation across datasets, hyperparameters, and out-of-distribution queries.

## 2 RELATED WORK

ANNS algorithms can be categorized into four fundamental paradigms by their exploration strategies, all leading to a refinement phase that requires exact distance evaluations.

**Graph-based methods** construct *navigable* networks, such as Navigable small World (NSW) (Malkov et al., 2014) graphs, built by greedy routing and enhanced via hierarchical layering to HNSW Malkov & Yashunin (2020). DiskANN (Subramanya et al., 2019) extends the idea to billion-scale datasets via disk-aware indexing with Vamana graphs (Subramanya et al., 2021). Sparse Navigable Graphs (SNG) (Khanna et al., 2025) utilize monotonic search paths, while Space Partition Tree And Graph (SPTAG) (Zhang et al., 2023) utilize relative neighborhood graphs and balanced $k$-means trees to facilitate the search. Still, these graph-based methods lead to a best-first search over candidate neighborhoods, which requires full distance evaluations.

**Clustering-and-quantization methods** partition space into regions and compress vector representations to reduce storage and accelerate computations through candidate filtering. IVF Jégou et al. (2008) creates inverted indices by $k$-means clustering and maps each query to its nearest cluster centroids to search within those. IVFPQ Jégou et al. (2011) combines clustering with *product quantization* to compress vector representations within clusters by decomposing them into subvectors and quantizing each independently. Advanced variants include OPQ Ge et al. (2014), which reduces quantization error via learned rotations, Multi-D-ADC, which assigns vectors to multiple clusters, and ScaNN Guo et al. (2020), which projects vectors to learned lower-dimensional subspaces by

anisotropic quantization. While these methods excel at compression and candidate reduction, they still require exact distance computations over quantized representations.

**Tree-based Methods** partition the vector space recursively and hierarchically. Classical k-d trees (Bentley, 1975) partition along coordinate axes, alternating dimensions per level, but degrade in high dimensions. BBF (bbf) improves k-d trees by searching bins in distance order. FLANN (Muja & Lowe, 2014) combines randomized k-d tree forests with hierarchical k-means clustering to create distribution-adaptive balanced partitions. Annoy Bernhardsson (2013) performs hyperplane splits optimized for memory mapping, while ball-trees (Omohundro, 1989) handle arbitrary distance functions through geometric properties by recursively partitioning data into nested hyperspheres. Still, tree-based methods eventually reach leaf nodes that require exact distance computations.

**Locality-Sensitive Hashing** (LSH) (Indyk & Motwani, 1998) maps nearby points to identical buckets with high probability. Extensions include $p$-stable distributions (Datar et al., 2004), E2LSH (Andoni & Indyk, 2006) with multiple hash tables, and multi-probe LSH (Lv et al., 2007), which examines neighboring buckets to reduce space requirements. Learning-based variants like C2LSH (Gionis et al., 1999) and QALSH (Huang et al., 2015) adapt hash functions to data distributions. Yet LSH methods also require exact distance verification over retrieved candidates.

**Probabilistic refinement** techniques such as FINGER (Chen et al., 2023), Probabilistic Kernel Methods (Lu et al., 2025), and the Bi-metric Framework (Xu et al., 2024) employ probabilistic bounds or proxy metrics, which allow for recall degradation and rely on graph-based indexes. Contrariwise, PANORAMA provides exact bounds, hence does not compromise recall, while being applicable to the refinement phase of any filtering-and-refinement algorithm.

Some works refer to the refinement step over $k' = k \cdot k_{\text{factor}}$ over-fetched candidates as **'reranking'**: IVFPQFastScan (André et al., 2015), which bucketizes codewords and computes SIMD-friendly bucket scores for vector encodings, and RaBitQ (Gao & Long, 2024), which uses bitwise encodings for faster distance computation. In contrast, *reranking* in Information Retrieval refers to reordering candidates by a different scoring model, such as a neural cross-encoder (Kurland & Lee, 2010). PANORAMA is naturally applicable to the refinement (or 'reranking') of over-fetched candidates.

## 3 PANORAMA: DISTANCE COMPUTATION

**Problem 1** (*k*NN refinement). *Given a query vector $\mathbf{q} \in \mathbb{R}^d$ and a candidate set $\Gamma = \{\mathbf{x}_1, \ldots, \mathbf{x}_N\}$, find the set $\mathcal{S} \subseteq \Gamma$ such that $|\mathcal{S}| = k$ and $\forall \mathbf{s} \in \mathcal{S}, \mathbf{x} \in \Gamma \setminus \mathcal{S} : \|\mathbf{q} - \mathbf{s}\|_2 \leq \|\mathbf{q} - \mathbf{x}\|_2$.*

**Problem 2** (ANN index). *An* approximate nearest neighbor *index is a function $\mathcal{I} : \mathbb{R}^d \times \mathbb{D} \to 2^{|\mathbb{D}|}$ that maps a query $\mathbf{q}$ and a set of vectors in a database $\mathbb{D}$ to a candidate set $\Gamma = \mathcal{I}(\mathbf{q}, \mathbb{D}) \subset \mathbb{D}$, where $\Gamma$ contains the true $k$-nearest neighbors with high probability.*[1]

Problem 1 poses a computational bottleneck: given $N$ candidates, naive refinement computes $\|\mathbf{q} - \mathbf{x}_i\|_2^2 = \sum_{j=1}^d (\mathbf{q}_j - \mathbf{x}_{i,j})^2$ for each $\mathbf{x}_i \in \Gamma$, requiring $\Theta(N \cdot d)$ operations.

Kashyap & Karras (2011) introduced STEPWISE $k$NN search, which incrementally incorporates features (i.e., dimensions) and refines lower (LB) and upper (UB) bounds for each candidate's distance from the query. This accretive refinement eventually yields exact distances. In addition, STEPWISE keeps track of the $k^{\text{th}}$ upper bound $d_k$ in each iteration, and prunes candidates having LB $> d_k$. When no more than $k$ candidates remain, these form the exact $k$NN result. We derive distance bounds using a norm-preserving transform $T : \mathbb{R}^d \to \mathbb{R}^d$ along the lines of (Kashyap & Karras, 2011), by decomposing the squared Euclidean distance as in:

$$\|\mathbf{q} - \mathbf{x}\|^2 = \|T(\mathbf{q})\|^2 + \|T(\mathbf{x})\|^2 - 2\langle T(\mathbf{q}), T(\mathbf{x})\rangle \tag{1}$$

Using thresholds $0 = m_0 < m_1 < \cdots < m_L = d$ partitioning vectors into $L$ levels $\ell_1, \ell_2, \ldots, \ell_L$, we define partial inner products and tail (*residual*) energies:

---

[1] Some indexes like HNSW perform filtering and refinement in tandem, thus not fitting our generalized definition of index; refining distances still takes up most of the query time.

$$p^{(\ell_1,\ell_2)}(\mathbf{q},\mathbf{x}) = \sum_{j=m_{\ell_1}+1}^{m_{\ell_2}} T(\mathbf{q})_j T(\mathbf{x})_j, \quad R_{T(\mathbf{q})}^{(\ell_1,\ell_2)} = \sum_{j=m_{\ell_1}+1}^{m_{\ell_2}} T(\mathbf{q})_j^2, \quad R_{T(\mathbf{x})}^{(\ell_1,\ell_2)} = \sum_{j=m_{\ell_1}+1}^{m_{\ell_2}} T(\mathbf{x})_j^2 \quad (2)$$

The inner product terms from level $m_\ell$ to the last dimension $d$ satisfy the Cauchy-Schwarz inequality (Horn & Johnson, 2012): $\left| \sum_{j=m_\ell+1}^{d} T(\mathbf{q})_j T(\mathbf{x})_j \right| \le \sqrt{R_{T(\mathbf{q})}^{(\ell,d)} R_{T(\mathbf{x})}^{(\ell,d)}}$, hence the bounds:

$$\mathsf{LB}^\ell(\mathbf{q},\mathbf{x}) = R_{T(\mathbf{q})}^{(0,d)} + R_{T(\mathbf{x})}^{(0,d)} - 2\left( p^{(0,\ell)}(\mathbf{q},\mathbf{x}) + \sqrt{R_{T(\mathbf{q})}^{(\ell,d)} R_{T(\mathbf{x})}^{(\ell,d)}} \right) \le \|\mathbf{q}-\mathbf{x}\|^2 \quad (3)$$

$$\mathsf{UB}^\ell(\mathbf{q},\mathbf{x}) = R_{T(\mathbf{q})}^{(0,d)} + R_{T(\mathbf{x})}^{(0,d)} - 2\left( p^{(0,\ell)}(\mathbf{q},\mathbf{x}) - \sqrt{R_{T(\mathbf{q})}^{(\ell,d)} R_{T(\mathbf{x})}^{(\ell,d)}} \right) \le \|\mathbf{q}-\mathbf{x}\|^2 \quad (4)$$

PANORAMA, outlined in Algorithm 1, maintains a heap $H$ of the exact $k$NN distances among processed candidates, initialized with the $k$ first read candidates, and the $k^{\text{th}}$ smallest distance $d_k$ from the query (Line 4). For subsequent candidates, it monotonically tightens the lower bound as $\mathsf{LB}^\ell(\mathbf{q},\mathbf{x}) \le \mathsf{LB}^{\ell+1}(\mathbf{q},\mathbf{x}) \le \|\mathbf{q} - \mathbf{x}\|^2$, and prunes the candidate once that lower bound exceeds the $d_k$ threshold (Line 8), enabling early termination at dimension $m_\ell < d$ (Line 9); otherwise, it reaches the exact distance and updates $H$ accordingly (Lines 12–14). Thanks

---

**Algorithm 1** PANORAMA: Iterative Distance Refinement

1: **Input:** Query $\mathbf{q}$, candidate set $\mathcal{C} = \{\mathbf{x}_1, \ldots, \mathbf{x}_{N'}\}$, transform $T$, $k$, batch size $B$
2: **Precompute:** $T(\mathbf{q})$, $\|T(\mathbf{q})\|^2$, and tail energies $R_q^{(\ell,d)}$ for all $\ell$
3: **Initialize:** Global exact distance heap $H$ (size $k$), global threshold $d_k \leftarrow +\infty$
4: Compute exact distances of first $k$ candidates, initialize $H$ and $d_k$
5: **for** each batch $\mathcal{B} \subset \mathcal{C}$ of size $B$ **do** $\quad \triangleright$ when $|\mathcal{B}| = 1$ the following reduces to each "**for** each candidate $\mathbf{x} \in \mathcal{C}$"
6: $\quad$ **for** $\ell = 1$ to $L$ **do**
7: $\quad\quad$ **for** each candidate $\mathbf{x} \in \mathcal{B}$ **do**
8: $\quad\quad\quad$ **if** $\mathsf{LB}^\ell(\mathbf{q},\mathbf{x}) > d_k$ **then** $\quad\quad\quad\quad\quad\quad\quad \triangleright$ Update LB bound
9: $\quad\quad\quad\quad$ Mark $\mathbf{x}$ as pruned $\quad\quad \triangleright$ If threshold exceeded, prune candidate
10: $\quad\quad\quad\quad$ **continue**
11: $\quad\quad\quad$ **if** $\pi = 1$ **then**
12: $\quad\quad\quad\quad$ Compute $\mathsf{UB}^\ell(\mathbf{q},\mathbf{x})$ $\quad\quad\quad\quad\quad\quad \triangleright$ Compute upper bound
13: $\quad\quad\quad\quad$ **if** $\mathsf{UB}^\ell(\mathbf{q},\mathbf{x}) < d_k$ **then**
14: $\quad\quad\quad\quad\quad$ Push $(\mathsf{UB}^\ell(\mathbf{q},\mathbf{x}), \mathbf{x})$ to $H$ as UB entry
15: $\quad\quad\quad\quad\quad$ Update $d_k = k^{\text{th}}$ distance in $H$; Crop $H$
16: $\quad$ **if** $\pi = 0$ **then**
17: $\quad\quad$ **for** each unpruned candidate $\mathbf{x} \in \mathcal{B}$ **do**
18: $\quad\quad\quad$ Push $(\mathsf{LB}^L(\mathbf{q},\mathbf{x}), \mathbf{x})$ to $H$ as exact entry $\triangleright \mathsf{LB}^L(\mathbf{q},\mathbf{x})$ is ED as $\ell = L$
19: $\quad\quad\quad$ **if** $d < d_k$ **then**
20: $\quad\quad\quad\quad$ Update $d_k = k^{\text{th}}$ distance in $H$; Crop $H$
21: **return** Candidates in $H$ (top $k$ with possible ties at $k^{\text{th}}$ position)

---

to the correctness of lower bounds and the fact that $d_k$ holds the $k^{\text{th}}$ distance among processed candidates, candidates that belong in the $k$NN result are not pruned. Algorithm 1 encapsulates a general procedure for several execution modes of PANORAMA. Appendix C provides further details on those modes. Notably, STEPWISE assumes a monolithic contiguous storage scheme, which does not accommodate the multifarious layouts used in popular ANNS indexes. We decouple the pruning strategy from memory layout with a *batch processing* framework that prescribes three execution modes using two parameters: a *batch size* $B$ and an *upper bound policy* $\pi \in \{0, 1\}$:

1. **Point-centric** ($B = 1, \pi = 0$), which processes candidates individually with *early abandoning*, hence suits graph- and tree-based indexes that store candidates in non-contiguous layouts.
2. **Batch-noUB** ($B > 1, \pi = 0$), which defers heap updates to reduce overhead and enhance throughput, appropriate for indexes organizing vectors in small batches.
3. **Batch-UB** ($B \gg 1, \pi = 1$), which amortizes system costs across large batches and uses upper bounds for fine-tuned pruning within each batch.

When using batches, we compute distances for candidates within a batch in tandem. Batch sizes are designed to fit in L1 cache and the additional cost is negligible. Section 6 provides more details.

**Theorem 1** (Computational Complexity). *Let $\rho_i \in \{m_0, \ldots, m_L\}$ be the dimension at which candidate $\mathbf{x}_i$ is pruned (or $d$ if $\mathbf{x}_i$ survives to the end). The total computational cost is $Cost = \sum_{i=1}^{N} \rho_i$, with expected cost $\mathbb{E}[Cost] = N\mathbb{E}[\rho]$. Defining $\phi = \frac{\mathbb{E}[\rho]}{d}$ as the average fraction of dimensions processed per candidate, the expected cost becomes $\mathbb{E}[Cost] = \phi \cdot d \cdot N$.*

PANORAMA relies on two design choices: first, a transform $T$ that concentrates energy in the leading dimensions, enabling tight bounds, which we achieve through *learned transforms* (Section 5) yielding exponential energy decay; second, *level thresholds* $m_\ell$ that strike a balance between the computational overhead level-wise processing incurs and the pruning granularity it provides.

## 4 THEORETICAL GUARANTEES

Here, we establish that, under a set of reasonable assumptions, the expected computational cost of PANORAMA significantly supersedes the brute-force approach. Our analysis is built on the pruning mechanism, the data distribution, and the properties of energy compaction, motivating our development of learned orthogonal transforms in Section 5. The complete proofs are in Appendix A.

**Notation.** We use asymptotic equivalence notation: for functions $f(n)$ and $g(n)$, we write $f(n) \sim c \cdot g(n)$ if $\lim_{n \to \infty} f(n)/g(n) = c$ for some constant $c > 0$. PANORAMA maintains a *pruning threshold* $d_k$ as the squared distance of the $k^{\text{th}}$ nearest neighbor among candidates processed so far. Candidates whose lower bound on distance exceeds this threshold are pruned. The pruning effectiveness depends on the *margin* $\Delta$ between a candidate's real distance and the threshold $d_k$. Larger margins allow for earlier pruning. Our analysis relies on the following assumptions:

A1. *Energy compaction:* we use an orthogonal transform $T$ that achieves *exponential* energy decay. The energy of vector $\mathbf{x}$ after the first $m$ dimensions is bounded by $R_{\mathbf{x}}^{(m,d)} \approx \|\mathbf{x}\|^2 e^{-\frac{\alpha m}{d}}$, where $\alpha > 1$ is an energy compaction parameter.

A2. *Level structure:* we use levels of a single dimension each, $m_\ell = \ell$, at the finest granularity.

A3. *Gaussian distance distribution:* the squared Euclidean distances of vectors from a given query $\mathbf{q}$, $\|\mathbf{q} - \mathbf{x}\|^2$, follow a Gaussian distribution.

A4. *Bounded norms:* all vectors have norms bounded by a constant $R$.

From these assumptions, we aggregate the cost of pruning over all candidates, analyzing the behavior of the margin $\Delta$ to derive the overall complexity. The full derivation in Appendix A provides a high-probability bound on the cost.

**Theorem 2** (Complexity). *By assumptions A1–A4, the expected computational cost to process a candidate set of size $N$ is:*

$$\mathbb{E}[Cost] \sim \frac{C \cdot Nd}{\alpha}$$

*where $C$ is a constant that approaches 1 as $N \to \infty$ under normalization.*

This result shows that any effective energy-compacting transform with $\alpha > 1$ strictly supersedes the naive complexity of $Nd$ (for which $C = 1$), while the compaction parameter $\alpha$ determines the speedup. Since $C \approx 1$ in practice (as confirmed by the empirical validation in Section 7.2), PANORAMA achieves an approximately $\alpha$-fold speedup. In effect, a larger $\alpha$ renders PANORAMA more efficient. We show that the analysis extends to the scenario of out-of-distribution (OOD) queries that do not compact as effectively as the database vectors:

**Theorem 3** (Robustness to Out-of-Distribution Queries). *Assume the query vector has energy compaction $\alpha_q$ and database vectors have compaction $\alpha_x$. Under assumptions A1–A4, the expected cost adheres to effective compaction $\alpha_{\text{eff}} = (\alpha_q + \alpha_x)/2$:*

$$\mathbb{E}[Cost] \sim \frac{C \cdot Nd}{\alpha_{\text{eff}}} \sim \frac{2C \cdot Nd}{\alpha_q + \alpha_x}$$

This result, shown in Section 7, demonstrates PANORAMA's robustness. Even if a query is fully OOD ($\alpha_q = 0$), the algorithm's complexity becomes $2C \cdot Nd/\alpha_x$, and still achieves a significant speedup provided the database is well-compacted, ensuring graceful performance degradation for challenging queries. In the following, we develop methods to learn data-driven orthogonal transforms that enhance energy compaction.

## 5 LEARNING ORTHOGONAL TRANSFORMS

Several linear orthogonal transforms, such as the Discrete Cosine Transform (DCT) and Discrete Haar Wavelet Transform (DHWT) Mallat (1999); Thomakos (2015), exploit local self-similarity properties in data arising from physical processes such as images and audio. However, these assumptions fail in modern high-dimensional machine learning datasets, e.g., word embeddings and document-term matrices. In these settings, classic transforms achieve limited energy compaction

and no permutation invariance. We address this deficiency via *transform learning*. Section 5.1 proposes a *learnable* linear orthogonal transform and Section 5.2 introduces a loss function tailored for ANNS-conductive energy compaction. Formally, we seek a matrix $T \in \mathbb{R}^{d \times d}$, with $T^\top T = I$, such that the transform $\mathbf{z} = T\mathbf{x}$ of a signal $\mathbf{x}$ attains *energy compaction*, i.e., concentrates most energy in its leading dimensions while preserving norms by orthogonality, i.e., $\|\mathbf{z}\|_2 = \|\mathbf{x}\|_2$.

## 5.1 ORTHOGONAL TRANSFORM PARAMETERIZATION

We view the set of orthogonal matrices, $\mathcal{O}(d) = \{T \in \mathbb{R}^{d \times d} : T^\top T = I\}$, as the *Stiefel manifold* (Edelman et al., 1998), a smooth Riemannian manifold where *geodesics* (i.e., straight paths on the manifold's surface) correspond to continuous rotations. The *Cayley transform* (Hadjidimos & Tzoumas, 2009; Absil et al., 2007) maps any $d \times d$ real skew-symmetric (antisymmetric) matrix $\mathbf{A}$— i.e., an element of the *Lie algebra* $\mathfrak{so}(d)$ of the special orthogonal group $\mathrm{SO}(d)$, with $\mathbf{A}^\top = -\mathbf{A}$, hence having $\dim = d(d-1)/2$ independent entries (Hall, 2013)—to an orthogonal matrix in $\mathrm{SO}(d)$ (excluding rotations with $-1$ eigenvalues). The resulting matrix lies on a subset of the Stiefel manifold, and the mapping serves as a smooth *retraction*, providing a first-order approximation of a geodesic at its starting point (Absil et al., 2007) while avoiding repeated projections:

$$T(\mathbf{A}) = \left(I - \tfrac{\gamma}{2}\mathbf{A}\right)^{-1}\left(I + \tfrac{\gamma}{2}\mathbf{A}\right). \tag{5}$$

The parameter $\gamma$ controls the step size of the rotation on the Stiefel manifold: smaller $\gamma$ values yield smaller steps, while larger values allow more aggressive rotations but may risk numerical instability. Contrary to other parameterizations for orthogonal transform operators, such as updates via *Householder reflections* Householder (1958) and *Givens rotations* Givens (1958), which apply a non-parallelizable sequence of simple rank-one or planar rotations, the Cayley map yields a full-matrix rotation in a single update step, enabling efficient learning on GPUs without ordering bias. Unlike *structured fast transforms* (Cooley & Tukey, 1965) (e.g., DCT), which rely on sparse, rigidly defined matrices crafted for specific data types, the learned transform is dense and fully determined by the data, naturally adapting to any dataset. Further, the Cayley map enables learning from a rich and continuous family of rotations; although it excludes rotations with $-1$ as an eigenvalue, which express a half-turn in some plane (Hall, 2013), it still allows gradient-based optimization using standard batched linear-algebra primitives, which confer numerical stability, parallelizability, and suitability for GPU acceleration.

## 5.2 ENERGY COMPACTION LOSS

As discussed, we prefer a transform that compacts the signal's energy into the leading dimensions and lets residual energies $R^{(\ell,d)}$ decay exponentially (Section 3). The *residual energy* of a signal $\mathbf{x}$ by an orthogonal transform $T$ following the first $\ell$ coefficients is $R_{T\mathbf{x}}^{(\ell,d)} = \sum_{j=\ell}^{d-1}(T\mathbf{x})_j^2$. We formulate a loss function that penalizes deviations of *normalized* residuals from exponential decay, on each dimension and for all vectors in a dataset $\mathcal{D}$, explicitly depending on the parameter matrix $\mathbf{A}$:

$$\mathcal{L}(T(\mathbf{A}); \mathcal{D}) = \frac{1}{N} \sum_{\mathbf{x} \in \mathcal{D}} \frac{1}{d} \sum_{\ell=0}^{d-1} \left( \frac{R_{T(\mathbf{A})\mathbf{x}}^{(\ell,d)}}{R_{T(\mathbf{A})\mathbf{x}}^{(0,d)}} - e^{-\frac{\alpha\ell}{d}} \right)^2, \quad \alpha > 0. \tag{6}$$

The learning objective is thus to find the optimal skew-symmetric matrix $\mathbf{A}^*$:

$$\mathbf{A}^* = \underset{\mathbf{A} \in \mathfrak{so}(d)}{\operatorname{argmin}} \mathcal{L}(T(\mathbf{A}); \mathcal{D}).$$

We target this objective by gradient descent, updating $\mathbf{A}$ at iteration $t$ as:

$$\mathbf{A}^{(t+1)} = \mathbf{A}^{(t)} - \eta\,\nabla_{\mathbf{A}}\mathcal{L}\big(T(\mathbf{A}^{(t)}); \mathcal{D}\big),$$

where $\eta$ is the learning rate, parameterizing only upper-triangular values of $\mathbf{A}$ to ensure it remains skew-symmetric. The process drives $\mathbf{A}$ in the skew-symmetric space so that the learned Cayley orthogonal transform $T(\mathbf{A}^{(t)})$, applied to the data in each step, compacts energy in the leading coefficients, leading residual energies $R_{T(\mathbf{A}^{(t)})\mathbf{x}}^{(\ell,d)}$ to decay quasi-exponentially. We set $\mathbf{A}^0 = 0^{(d \times d)}$, hence $T(\mathbf{A}^0) = I$, and *warm-start* by composing it with the orthogonal PCA basis $T'$, which projects energy to leading dimensions (Yang et al., 2025). The initial transform is thus $T'$, and subsequent gradient updates of $\mathbf{A}$ adapt the composed orthogonal operator $T(\mathbf{A})T'$ to the data.

# 6 INTEGRATION WITH STATE-OF-THE-ART INDEXES

State-of-the-art ANNS indexes fall into two categories of memory layout: *contiguous*, which store vectors (or codes) consecutively in memory, and *non-contiguous*, which scatter vectors across non-consecutive locations (Han et al., 2023). On contiguous layouts, which exploit spatial locality and SIMD parallelism, we rearrange the contiguous storage to a *level-major* format to facilitate PANORAMA's *level-wise* refinement and bulk pruning in cache-efficient fashion. On non-contiguous layouts, PANORAMA still curtails redundant distance computations, despite the poor locality. Here, we discuss how we integrate PANORAMA in the refinement step of both categories.

## 6.1 CONTIGUOUS-LAYOUT INDEXES

**L2Flat and IVFFlat.** L2Flat (Douze et al., 2024) (Faiss's naive $k$NN implementation) performs a brute-force $k$NN search over the entire dataset. IVFFlat (Jégou et al., 2008) implements *inverted file indexing*: it partitions the dataset into $n_{\text{list}}$ clusters by $k$-means and performs a brute-force $k$NN over the points falling within the nearest $n_{\text{probe}}$ clusters to the query point. Nonetheless, their native storage layout does not suit PANORAMA for two reasons:

1. **Processor cache locality and prefetching:** By PANORAMA refinement, we reload query slices for each vector, preventing stride-based prefetching and causing frequent processor cache misses.

2. **Branch misprediction:** While processing a single vector, the algorithm makes up to $n_{\text{levels}}$ decisions on whether to prune it, each introducing a branch, which renders control flow irregular, defeats branch predictors, and stalls the instruction pipeline.

To address these concerns, we integrate PANORAMA in Faiss (Douze et al., 2024) with a *batched, level-major design*, restructuring each cluster's memory layout to support *level-wise* (i.e., one level at a time) rather than *vector-wise* refinement. We group vectors into *batches* and organize each batch in *level-major* order that generalizes the *dimension-major* layout of PDX (Kuffo et al., 2025). Each *level* stores a contiguous group of features for each point in the batch. The algorithm refines distances level-by-level within each batch. At each level, it first computes the distance contributions for all vectors

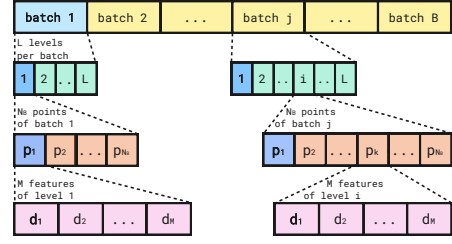

Figure 3: IVFFlat & L2Flat storage.

in the batch, and then makes *bulk pruning* decisions over all vectors. This consolidation of branch checks in $n_{\text{levels}}$ synchronized steps regularizes control flow, reduces branch mispredictions, and improves cache utilization (Ailamaki et al., 2001). Figure 3 illustrates the principle.

**IVFPQ.** (Jégou et al., 2011) combines *inverted file indexing* with *product quantization* (PQ) to reduce memory usage. It first assigns a query to a coarse cluster (as in IVFFlat), and then approximates distances within that cluster using PQ-encoded vectors (*codes*): it divides each $d$-dimensional vector into $M$ contiguous subvectors of size $d/M$, applies $k$-means in each subvector space separately to learn $2^{n_{\text{bits}}}$ centroids, and compactly represents each subvector using $n_{\text{bits}}$ bits. However, directly applying the storage layout of Figure 3 to quantization codes introduces an additional challenge:

3. **SIMD lane underutilization:** When the PQ codes for a given vector are shorter than the SIMD register width, vector-wise processing leaves many lanes idle, underusing compute resources.

Instead of storing PQ codes by vector, we contiguously store code slices of the same quantizer across vectors in a batch as Figure 4 depicts. This layout lets SIMD instructions process lookup-table (LUT) entries for multiple vectors in parallel within the register, fully utilizing compute lanes (Li & Patel, 2013; Feng et al., 2015), and reduces cache thrashing, as LUT entries of codes for the same query slices remain cache-resident for reuse. We evaluate this effect, along with varying level settings, in Appendix F.6.

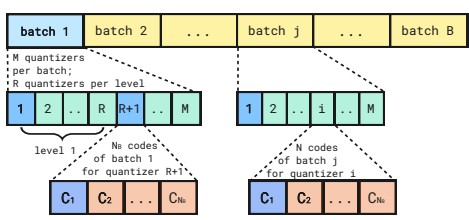

Figure 4: IVFPQ; codes absorb dimensions.

## 6.2 Non-contiguous-layout indexes

On index methods that store candidate points in noncontiguous memory, the refinement phase faces a memory–computation tradeoff. Fetching candidate vectors incurs frequent processor (L3) cache misses, so the cost of moving data into cache rivals that of arithmetic distance computations, rendering the process *memory-bound*. Even with SIMD acceleration, poor locality among candidates slows throughput, and by Amdahl's law (1967), enhancing computation alone yields diminishing returns. Lacking a good fix, we do not rearrange the storage layout with these three indexes.

**Graph-based (HNSW).** HNSW (Malkov & Yashunin, 2020) organizes points in a multi-layer graph, reminiscent of a skip list; upper layers provide logarithmic long-range routing while lower layers ensure local connectivity. To navigate this graph efficiently, it prioritizes exploration using a *candidate heap* and organizes $k$NN results using a *result heap*. Unlike other ANNS methods, HNSW conducts no separate verification, as it computes exact distances during traversal. We integrate PANORAMA by modifying how embeddings enter the candidate heap to reduce distance evaluations: we prioritize candidates using running distance bounds, with the estimate $\frac{\mathsf{LB}^\ell + \mathsf{UB}^\ell}{2}$, and defer computing a candidate's exact distance until it enters the result heap.

**Tree-based (Annoy).** Tree-based methods recursively partition the vector space into leaf nodes, each containing candidate vectors. Annoy (Bernhardsson, 2013) constructs these partitions by splitting along hyperplanes defined by pairs of randomly selected vectors, and repeats this process to build a *random forest* of $n_{\text{trees}}$ trees. At query time, it traverses each tree down to the nearest leaf and sends the candidate vectors from all visited leaves to verification, where we integrate PANORAMA.

**Locality-based (MRPT).** MRPT (Multiple Random Projection Trees) (Hyvönen et al., 2016; Hyvönen et al., 2016; Jääsaari et al., 2019a) also uses a forest of random trees, like Annoy does, yet splits via median thresholds on *random linear projections* rather than via hyperplanes. This design ties MRPT to Johnson–Lindenstrauss guarantees, enabling recall tuning, and incorporates voting across trees to filter candidates. We integrate PANORAMA as-is in the refinement phase.

## 6.3 Memory Footprint

To apply the Cauchy-Schwarz bound approximation, we precompute tail energies of transformed vectors at each level, with an $O(nL)$ memory overhead, where $n$ is the dataset size and $L$ the number of levels. For IVFPQ using $M = 480$ subquantizers on GIST, $n_{\text{bits}} = 8$ bits per code, and $L = 8$ levels at 90% recall, this results in a 7.5% additional storage overhead. On methods that do not quantize vectors, the overhead is even smaller (e.g., 0.94% in IVFFlat). In addition, we incur a small fixed-size overhead to store partial distances in a batch, which we set to fit within L1 cache.

## 7 Experimental Results

We comprehensively evaluate PANORAMA's performance in terms of the speedup it yields when integrated into existing ANNS methods, across multiple datasets.[2]

**Datasets.** Table 1 lists our datasets. **CIFAR-10** contains flattened natural-image pixel intensities. **FashionMNIST** provides representations of grayscale clothing items. **GIST** comprises natural scene descriptors. **SIFT** provides scale-invariant feature transform descriptors extracted from images. **DBpedia-Ada** (Ada) holds OpenAI's `text-embedding-ada-002` representations of DBpedia entities, a widely used semantic-search embedding model, and **DBpedia-Large** (Large) lists higher-dimensional embeddings of the same corpus by `text-embedding-3-large`.

Table 1: Data extents.

| Data | $n$ | $d$ |
|---|---|---|
| SIFT | 10M/100M | 128 |
| GIST | 1M | 960 |
| FashionMNIST | 60K | 784 |
| Ada | 1M | 1536 |
| Large | 1M | 3072 |
| CIFAR-10 | 50K | 3072 |

---

[2]Experiments were conducted on an `m6i.metal` Amazon EC2 instance with an Intel(R) Xeon(R) Platinum 8375C CPU @2.90GHz and 512GB of DDR4 RAM running Ubuntu 24.04.3 LTS. All binaries were compiled with GCC 13.3.0, enabled with AVX-512 flags up to VBMI2 and $-$O3 optimizations. The code is available at https://anonymous.4open.science/r/panorama-iclr.

**Methodology.** First, we measure PANORAMA's gains over Faiss' brute-force $k$NN implementation. Second, we gauge the gain of integrating PANORAMA into state-of-the-art ANNS methods. Third, we assess robustness under out-of-distribution queries. For each measurement, we run 5 repetitions of 100 10NN queries randomly selected from the benchmark query set and report averages.

## 7.1 FUNDAMENTAL PERFORMANCE ON LINEAR SCAN

Here, we measure speedups on a naive linear scan (Faiss' L2Flat) to assess our approach without integration complexities. We compute speedup by running 5 runs of 100 queries, averaging queries per second (QPS) across runs. Figure 5 plots our results, with *speedup* defined as $\mathrm{QPS_{Panorama}}/\mathrm{QPS_{L2Flat}}$. Each bar shows a speedup value and whiskers indicate standard deviations, estimated by the delta method, assuming independence between the two QPS values: $\sigma_S \approx \sqrt{\sigma_X^2/\mu_Y^2 + \mu_X^2 \sigma_Y^2/\mu_Y^4}$, where $\mu_X, \sigma_X$ are

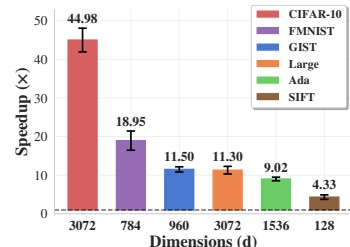

Figure 5: Speedups on $k$NN.

the mean and standard deviation of $\mathrm{QPS_{Panorama}}$, and $\mu_Y, \sigma_Y$ of $\mathrm{QPS_{L2Flat}}$. Each bar is capped with the value of $\mu_X/\mu_Y$. PANORAMA achieves substantial acceleration across datasets, while the high-dimensional CIFAR-10 data achieves the highest speedup, validating our predictions.

## 7.2 ENERGY COMPACTION

We gauge the energy compaction by our learned transforms $T \in \mathcal{O}(d)$, via normalized tail energies $\bar{R}^{(\ell,d)} = \frac{R^{(\ell,d)}}{R^{(0,d)}}$. An apt transform should gather energy in the leading dimensions, causing $\bar{R}^{(\ell,d)}$

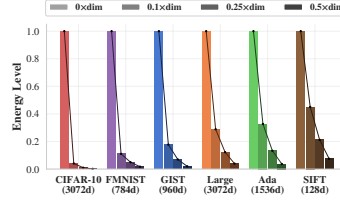

Figure 6: Energy compaction.

Table 2: Processed features.

| Dataset | Expected (%) | Empirical (%) |
|---|---|---|
| Large | 8.96 | 8.22 |
| Ada | 8.06 | 8.21 |
| FashionMNIST | 4.54 | 6.75 |
| GIST | 5.78 | 4.28 |
| CIFAR-10 | 3.12 | 3.71 |
| SIFT | 12.54 | 12.76 |

to decay rapidly. Figure 6 traces this decay across datasets for $p = \frac{\ell}{d} \in \{0, 0.1, 0.25, 0.5\}$. A steep decline indicates energy compaction aligned with the target. We also estimate the compaction parameter $\alpha$ from measured energies for $p = \frac{\ell}{d} \in \{0.1, 0.25, 0.5\}$ as $\alpha_p = -\frac{1}{p} \ln \frac{R^{(pd,d)}}{R^{(0,d)}}$, and average across $p$ for stability. By Theorem 2, the expected ratio of features processed before pruning a candidate is $\mathbb{E}[d_i] \propto d/\alpha$. Table 2 reports expected ratios (in %) alongside average empirical values. Their close match indicates that PANORAMA achieves the expected $\alpha$-fold speedup, hence $C \approx 1$ in Theorem 2.

## 7.3 INTEGRATION WITH ANN INDICES

We now assess PANORAMA's integration with ANN indices. Figure 7 plots $\frac{\mathrm{QPS_{Index+Panorama}}}{\mathrm{QPS_{Index}}}$ vs. recall. We collect recall–QPS pairs via a hyperparameter scan on the base index as in Figure 18. **IVFFlat** exhibits dramatic speedups of 2-40×, thanks to contiguous memory access. **IVFPQ** shows speedups of 2–30×, particularly at high recall levels where large candidate sets admit effective pruning. As product quantization does not preserve norms, the recall of PANORAMA IVFPQ, which applies PQ on transformed data, differs from that of the standard version for the same setting. We thus interpolate recall-QPS curves to compute speedup. **HNSW** presents improvements of up to 4×, despite the complexity of graph traversal. While tree-based **Annoy** and **MRPT** spend less time in verification compared to IVFPQ and IVFFlat as Figure 2 shows, we still observe gains of up to 6×.

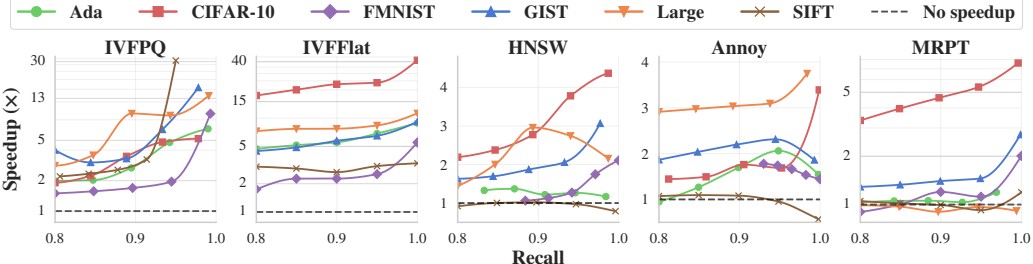

Figure 7: Speedup vs. recall. SIFT-10M data with HNSW, Annoy, MRPT; SIFT-100M with others.

### 7.4 CONTRIBUTION OF THE TRANSFORM

Here, we study the individual contributions of PANORAMA's bounding methodology and of its learned orthogonal transforms. We apply PANORAMA with all ANNS indices on the GIST1M dataset in two regimes: (i) on original data vectors, and (ii) on vectors transformed by the learned energy-compacting transform. Figure 8 presents the results, plotting speedup over the baseline index vs. recall. While PANORAMA on original data accelerates search thanks to partial-product pruning, the transform consistently boosts these gains, as it tightens the Cauchy–Schwarz bounds.

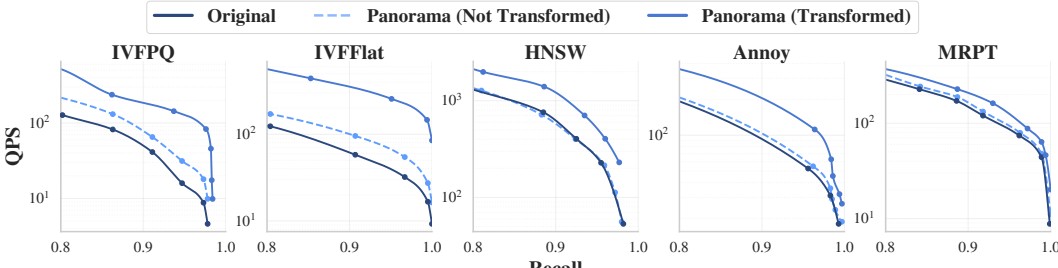

Figure 8: Speedup on GIST1M: PANORAMA on original vs. transformed data.

### 7.5 OUT-OF-DISTRIBUTION QUERY ANALYSIS

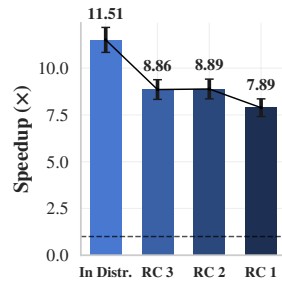

Figure 9: Query hardness.

To assess PANORAMA's robustness, we use synthetic out-of-distribution (OOD) queries crafted by *Hephaestus* (Ceccarello et al., 2025), which controls query difficulty by *Relative Contrast* (RC)—the ratio between the average distance from a query $\mathbf{q}$ to points in dataset $S$ and the distance to its $k^{\text{th}}$ nearest neighbor: $RC_k(\mathbf{q}) = \frac{1}{|S|}\sum_{x \in S} d(\mathbf{q}, x)/d(\mathbf{q}, x^{(k)})$. Smaller RC values indicate harder queries. We experiment with OOD queries of RC values of 3 (easy), 2 (medium), and 1 (hard) on the GIST1M dataset, computed with respect to 10 nearest neighbors. Figure 9 plots PANORAMA's performance under OOD queries. Although OOD queries may exhibit poor energy compaction by the learned transform, PANORAMA attains robust speedup thanks to the structure of Cauchy-Schwarz bounds. By Equation (2), pruning relies on the product of database and query energies, $R_{T(\mathbf{x})}$ and $R_{T(\mathbf{q})}$. Well-compacted database vectors couteract poor query compaction, so the geometric mean $\sqrt{R_{T(\mathbf{q})}R_{T(\mathbf{x})}}$ bound remains effective. Theorem 8 supports this conclusion. Observed speedups thus align with theory across RC levels.

### 7.6 ADDITIONAL EXPERIMENTS

We conduct comprehensive ablation studies to further validate PANORAMA's design choices and system implementation. Our ablations demonstrate that PANORAMA's adaptive pruning significantly outperforms naive dimension truncation approaches, which suffer substantial recall degradation. We compare using PCA and DCT methods against learned Cayley transforms. Systematic analysis reveals that PANORAMA's performance scales favorably and as expected with dataset size, dimensionality and $k$. We identify optimal configurations for the number of refinement levels and show that measured speedups align with expected performance from our system optimizations. Complete experimental details are provided in Appendix F.

## 8 CONCLUSION

We proposed PANORAMA, a theoretically justified fast-track technique for the refinement phase in production ANNS systems, leveraging a data-adaptive learned orthogonal transform that compacts signal energy in the leading dimensions and a bounding scheme that enables candidate pruning with partial distance computations. We integrate PANORAMA into contiguous-layout and non-contiguous-layout ANNS indexes, crafting tailored memory layouts for the former that allow full SIMD and cache utilization. Our experiments demonstrate PANORAMA to be viable and effective, scalable to millions of vectors, and robust under challenging out-of-distribution queries, attaining consistent speedups while maintaining search quality.

## 9 REPRODUCIBILITY STATEMENT

To ensure reproducibility, we provide several resources alongside this paper. Our source code and implementations are publicly available at `https://anonymous.4open.science/r/panorama-iclr`, including scripts for integrating PANORAMA with baseline indexes and reproducing all results. Appendix A contains full proofs of all theoretical results and assumptions, ensuring clarity in our claims. Appendix B documents the complete experimental setup, including hardware/software specifications, datasets, parameter grids, and training details. Additional implementation notes, integration details, and extended ablations are provided in Appendices C–F.

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

## APPENDIX LAYOUT

This appendix complements the main text with detailed proofs, algorithmic insights, implementation notes, and extended experiments.

1. **Proofs** (Appendix A): Full, formal proofs for all theorems, lemmas, and claims stated in the main text. Each proof is cross-referenced to the corresponding result in the paper, and we include any auxiliary lemmas and technical bounds used in the derivations.

2. **Experimental setup** (Appendix B): Complete experimental details necessary for reproducibility, including dataset descriptions, evaluation metrics, hyperparameter grids, indexing parameters (e.g., $n_{\text{list}}$, $n_{\text{probe}}$, $ef_{\text{search}}$), hardware/software environment.

3. **Panorama details** (Appendix C): Expanded algorithmic description of PANORAMA, with full pseudocode for all variants, implementation notes, complexity discussion, and additional examples illustrating batching, and level-major ordering.

4. **HNSW** (Appendix D): Non-trivial integration of PANORAMA with HNSW. Contains the HNSW+Panorama pseudocode, correctness remarks, and heuristics for beam ordering with heterogeneous (partial/exact) distance estimates.

5. **Systems details** (Appendix E): Low-level implementation details pertaining to IVFPQ. This section documents our PANORAMA integration into Faiss, detailing buffering and scanning strategies for efficient SIMD vectorization.

6. **Ablations** (Appendix F): Extended ablation studies and plots not included in the main body, including per-dataset and per-index breakdowns, PCA/DCT/Cayley comparisons, scaling with $N, d, k$, and comparisons between expected and measured speedups.

## A    THEORETICAL ANALYSIS OF PANORAMA'S COMPLEXITY

This appendix derives the expected computational complexity of the Panorama algorithm. The proof proceeds in six steps, starting with a statistical model of the candidate distances and culminating in a final, simplified complexity expression.

**Notation.**    Throughout this analysis, we use asymptotic equivalence notation: for functions $f(n)$ and $g(n)$, we write $f(n) \sim c \cdot g(n)$ if $\lim_{n\to\infty} f(n)/g(n) = c$ for some constant $c > 0$. When $c = 1$, we simply write $f(n) \sim g(n)$.

### SETUP AND ASSUMPTIONS

Our analysis relies on the following assumptions:

- **(A1) Optimal Energy Compaction:** A learned orthogonal transform $T$ is applied, such that the tail energy of any vector $\mathbf{v}$ decays exponentially: $R_{\mathbf{v}}^{(m,d)} := \sum_{j=m+1}^{d} T_j(\mathbf{v})^2 \approx \|\mathbf{v}\|^2 e^{-\alpha m/d}$, where $\alpha$ is the energy compaction parameter.

- **(A2) Level Structure:** We use single-dimension levels for the finest pruning granularity: $m_\ell = \ell$.

- **(A3) Gaussian Approximation of Distance Distribution:** The squared Euclidean distances, $\|\mathbf{q} - \mathbf{x}_i\|^2$, are modeled using a Gaussian approximation (e.g., via the central limit theorem for large $d$) with mean $\mu$ and standard deviation $\sigma$. The exact distribution is chi-square-like; we use the Gaussian for tractability.

- **(A4) Bounded Norms:** Vector norms are uniformly bounded: $\|\mathbf{q}\|, \|\mathbf{x}_i\| \leq R$ for some constant $R$.

### STEP 1: MARGIN DEFINITION FROM SAMPLED-SET STATISTICS

The Panorama algorithm (Algorithm 4) maintains a pruning threshold $\tau$, which is the squared distance of the $k$-th nearest neighbor found so far. For analytical tractability, we model $\tau_i$ as the $k$-th order statistic among $i$ i.i.d. draws from the distance distribution, acknowledging that the algorithm's threshold arises from a mixture of exact and pruned candidates. We begin by deriving a high-probability bound on this threshold after $i$ candidates have been processed.

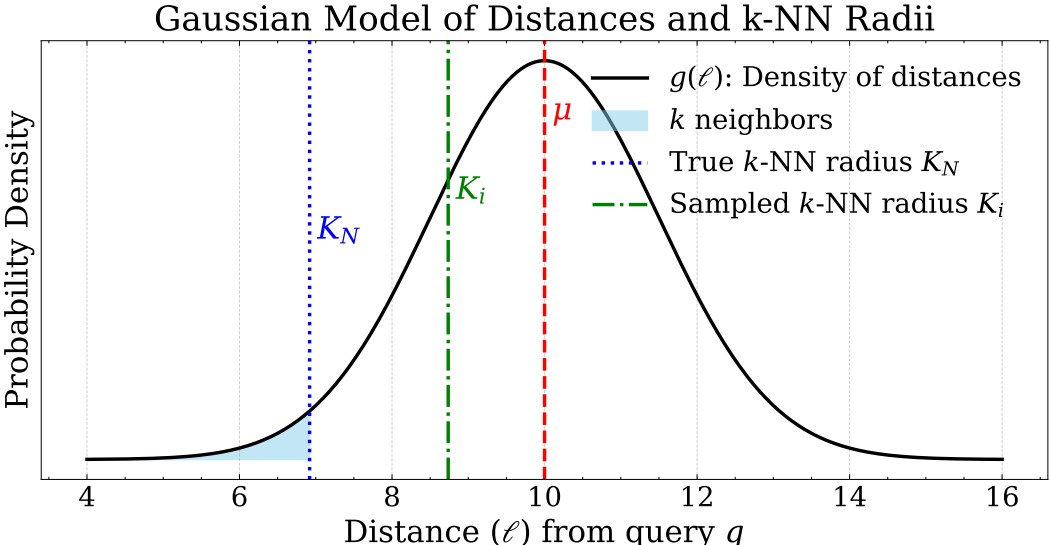

Figure 10: Visualization under a Gaussian approximation of the distance distribution. The curve represents the probability density of squared distances from a query **q**. $\mu$ is the mean distance. For a full dataset of $N$ points, the $k$-NN distance threshold is $K_N$, enclosing $k$ points. When we take a smaller candidate sample of size $i < N$, the expected $k$-NN threshold, $K_i$, is larger than $K_N$. The margin for a new candidate is its expected distance ($\mu$) minus this sampled threshold $K_i$.

**Theorem 4** (High-probability bound for the sampled k-NN threshold via DKW). *Let the squared distances be i.i.d. random variables with CDF $F(r)$. For any $\varepsilon \in (0,1)$, with probability at least $1 - 2e^{-2i\varepsilon^2}$ by the Dvoretzky–Kiefer–Wolfowitz inequality Wikipedia contributors (2025); Massart (1990), the $k$-th order statistic $\tau_i$ satisfies*

$$F^{-1}\left(\max\left\{0, \tfrac{k}{i+1} - \varepsilon\right\}\right) \;\le\; \tau_i \;\le\; F^{-1}\left(\min\left\{1, \tfrac{k}{i+1} + \varepsilon\right\}\right).$$

*Under the Gaussian assumption (A3), where $F(r) = \Phi\left(\frac{r-\mu}{\sigma}\right)$, this implies in particular the upper bound*

$$\tau_i \;\le\; \mu + \sigma\,\Phi^{-1}\left(\tfrac{k}{i+1} + \varepsilon\right) \quad \text{with probability at least } 1 - 2e^{-2i\varepsilon^2}.$$

*Proof.* Let $F_i$ be the empirical CDF of the first $i$ distances. The DKW inequality gives $\Pr\left(\sup_t |F_i(t) - F(t)| > \varepsilon\right) \le 2e^{-2i\varepsilon^2}$ Massart (1990). On the event $\sup_t |F_i - F| \le \varepsilon$, we have for all $t$: $F(t) - \varepsilon \le F_i(t) \le F(t) + \varepsilon$. Monotonicity of $F^{-1}$ implies $F^{-1}(u - \varepsilon) \le F_i^{-1}(u) \le F^{-1}(u + \varepsilon)$ for all $u \in (0,1)$. Taking $u = k/(i+1)$ and recalling that $\tau_i = F_i^{-1}(k/(i+1))$ yields the two-sided bound. Under (A3), $F^{-1}(p) = \mu + \sigma\,\Phi^{-1}(p)$, which gives the Gaussian form. $\square$

A new candidate is tested against this threshold $\tau_i$. Its expected squared distance is $\mu$. This allows us to define a high-probability margin.

**Definition 1** (High-probability Margin $\Delta_i$). *Fix a choice $\varepsilon_i \in (0,1)$. Define the sampled k-NN threshold upper bound*

$$K_i := F^{-1}\left(\tfrac{k}{i+1} + \varepsilon_i\right) = \mu + \sigma\,\Phi^{-1}\left(\tfrac{k}{i+1} + \varepsilon_i\right).$$

*Then define the margin as*

$$\Delta_i := \mu - K_i = -\sigma\,\Phi^{-1}\left(\tfrac{k}{i+1} + \varepsilon_i\right).$$

*With probability at least $1 - 2e^{-2i\varepsilon_i^2}$, a typical candidate with expected squared distance $\mu$ has margin at least $\Delta_i$. For this margin to be positive (enabling pruning), it suffices that $\frac{k}{i+1} + \varepsilon_i < 0.5$ (equivalently, $\Phi^{-1}(\cdot) < 0$). In what follows in this section, interpret $\Delta_i$ as this high-probability margin so that subsequent bounds inherit the same probability guarantee (optionally uniformly over $i$ via a union bound).*

**Uniform high-probability schedule.** Fix a target failure probability $\delta \in (0,1)$ and define

$$\varepsilon_i := \sqrt{\frac{1}{2i} \log\left(\frac{2N'}{\delta}\right)}.$$

By a union bound over $i \in \{k+1, \ldots, N'\}$, the event

$$\mathcal{E}_\delta := \bigcap_{i=k+1}^{N'} \left\{ \tau_i \leq \mu + \sigma\, \Phi^{-1}\left(\frac{k}{i+1} + \varepsilon_i\right) \right\}$$

holds with probability at least $1 - \delta$. All bounds below are stated on $\mathcal{E}_\delta$.

STEP 2: PRUNING DIMENSION FOR A SINGLE CANDIDATE

A candidate $\mathbf{x}_j$ is pruned at dimension $m$ if its lower bound exceeds the threshold $\tau$. A sufficient condition for pruning is when the worst-case error of the lower bound is smaller than the margin (for the candidate processed at step $i$):

$$\|\mathbf{q} - \mathbf{x}_j\|^2 - \mathrm{LB}^{(m)}(\mathbf{q}, \mathbf{x}_j) < \Delta_i$$

From the lower bound definition in Equation (3), the error term on the left is bounded by four times the geometric mean of the tail energies in the worst case. Applying assumption (A1) for energy decay and (A4) for bounded norms, we get:

$$4\sqrt{R_{\mathbf{q}}^{(m,d)} R_{\mathbf{x}_j}^{(m,d)}} \leq 4\sqrt{(\|\mathbf{q}\|^2 e^{-\alpha m/d})(\|\mathbf{x}_j\|^2 e^{-\alpha m/d})} \leq C_0\, e^{-\alpha m/d}$$

Here and henceforth, let $C_0 := 4R^2$. The pruning condition thus becomes:

$$C_0\, e^{-\alpha m/d} < \Delta_i$$

We now solve for $m$, which we denote the pruning dimension $d_j$:

$$e^{-\alpha d_j/d} < \frac{\Delta_i}{C_0}$$

$$-\frac{\alpha d_j}{d} < \log\left(\frac{\Delta_i}{C_0}\right)$$

$$\frac{\alpha d_j}{d} > -\log\left(\frac{\Delta_i}{C_0}\right) = \log\left(\frac{C_0}{\Delta_i}\right)$$

$$d_j > \frac{d}{\alpha} \log\left(\frac{C_0}{\Delta_i}\right)$$

**Theorem 5** (Pruning dimension $d_i$). *The expected number of dimensions $d_i$ processed for a candidate at step $i$ is approximately:*

$$d_i \approx \frac{d}{\alpha} \left[\log\left(\frac{C_0}{\Delta_i}\right)\right]_+$$

*where $C_0 = 4R^2$ encapsulates the norm-dependent terms and $[x]_+ := \max\{0, x\}$.*

STEP 3: TOTAL COMPUTATIONAL COMPLEXITY

The total computational cost of Panorama is dominated by the sum of the pruning dimensions for all $N'$ candidates in the candidate set $\mathcal{C}$. Define the first index at which the high-probability margin becomes positive as

$$i_0 := \min\left\{i \geq k+1 : \frac{k}{i+1} + \varepsilon_i < \frac{1}{2}\right\}.$$

Then

$$\text{Cost} = \sum_{i=k+1}^{N'} d_i \approx \sum_{i=\max\{i_0,\, k+1\}}^{N'} \frac{d}{\alpha} \left[\log\left(\frac{C_0}{\Delta_i}\right)\right]_+$$

Let $I_{C_0} := \{i \in \{\max\{i_0,\, k+1\}, \ldots, N'\} : \Delta_i \leq C_0\}$. Denote by $N'_{C_0} := \max I_{C_0}$ the largest contributing index. Then

$$\sum_{i=k+1}^{N'} \left[ \log\left( \frac{C_0}{\Delta_i} \right) \right]_+ = \sum_{i \in I_{C_0}} (\log C_0 - \log \Delta_i)$$

$$= |I_{C_0}| \log C_0 - \log\left( \prod_{i \in I_{C_0}} \Delta_i \right)$$

**Theorem 6** (Complexity via margin product). *The total computational cost is given by:*

$$Cost \approx \frac{d}{\alpha} \left( |I_{C_0}| \log C_0 - \log\left( \prod_{i \in I_{C_0}} \Delta_i \right) \right)$$

STEP 4: ASYMPTOTIC ANALYSIS OF THE MARGIN PRODUCT

To evaluate the complexity, we need to analyze the product of the margins over the contributing indices, $P = \prod_{i \in I_{C_0}} \Delta_i$. We use the well-known asymptotic for the inverse normal CDF for small arguments $p \to 0$: $\Phi^{-1}(p) \sim -\sqrt{2 \ln(1/p)}$. In our case, for large $i$, $p = \frac{k}{i+1} + \varepsilon_i$ is small provided $\varepsilon_i = o(1)$.

$$\Delta_i = -\sigma\, \Phi^{-1}\left( \tfrac{k}{i+1} + \varepsilon_i \right) \approx \sigma \sqrt{2 \ln\left( \frac{i+1}{k + (i+1)\varepsilon_i} \right)}$$

The logarithm of the product is the sum of logarithms. Note the sum starts from $i = i_0$ (the first index where $\Delta_i > 0$), and is further truncated at the largest index $N'_{C_0}$ for which $\Delta_i \leq C_0$.

$$\log(P) = \sum_{i=i_0}^{N'_{C_0}} \ln(\Delta_i) \approx \sum_{i=i_0}^{N'_{C_0}} \left[ \ln \sigma + \frac{1}{2} \ln\left( 2\, \ln\left( \frac{i}{k + i\varepsilon_i} \right) \right) \right]$$

For large $N'_{C_0}$, the term $\ln(\ln(\frac{i}{k+i\varepsilon_i}))$ changes very slowly. The following bound formalizes this heuristic.

**Lemma 1** (Bounding the slowly varying sum). *Let $g(i) := \ln\big( \ln(i/(k + i\varepsilon_i)) \big)$ for $i \geq i_0$, where $\varepsilon_i$ is nonincreasing. Then for any integers $a < b$,*

$$\sum_{i=a}^{b} g(i) \leq (b - a + 1)\, g(b) + \int_a^b \frac{1}{x\, \ln(x/(k + x\varepsilon_x))}\, dx.$$

*In particular, taking $a = i_0$ and $b = N'_{C_0}$ and noting that the integral term is bounded by an absolute constant multiple of $\ln\ln\big( N'_{C_0}/(k + N'_{C_0} \varepsilon_{N'_{C_0}}) \big)$, we obtain*

$$\sum_{i=i_0}^{N'_{C_0}} \ln\left( \ln\left( \frac{i}{k + i\varepsilon_i} \right) \right) \leq (N'_{C_0} - i_0 + 1)\, \ln\left( \ln\left( \frac{N'_{C_0}}{k + N'_{C_0} \varepsilon_{N'_{C_0}}} \right) \right) + c_0\, \ln\ln\left( \frac{N'_{C_0}}{k + N'_{C_0} \varepsilon_{N'_{C_0}}} \right)$$

*for some absolute constant $c_0 > 0$.*

Applying this lemma to $\log(P)$ yields the explicit bound

$$\log(P) \leq (N'_{C_0} - i_0 + 1) \left( \ln \sigma + \frac{1}{2} \ln\left( 2\, \ln\left( \frac{N'_{C_0}}{k + N'_{C_0} \varepsilon_{N'_{C_0}}} \right) \right) \right) + c_0\, \ln\ln\left( \frac{N'_{C_0}}{k + N'_{C_0} \varepsilon_{N'_{C_0}}} \right).$$

STEP 5: FINAL COMPLEXITY RESULT

Substituting the asymptotic result for the margin product with high-probability margins back into our complexity formula, we arrive at the final statement (holding with probability at least $1 - \sum_i 2e^{-2i\varepsilon_i^2}$ if a union bound over $i$ is applied).

**Theorem 7** (Final complexity of Panorama). *The expected computational cost to process a candidate set is:*

$$\mathbb{E}[Cost] \approx \frac{d}{\alpha}\left(|I_{C_0}|\log C_0 - (N'_{C_0} - i_0 + 1)\left[\ln\sigma + \frac{1}{2}\ln\left(2\,\ln\left(\frac{N'_{C_0}}{k + N'_{C_0}\varepsilon_{N'_{C_0}}}\right)\right)\right]\right)$$

STEP 6: FINITE-SAMPLE BOUND

On the event $\mathcal{E}_\delta$ (which holds with probability at least $1 - \delta$), combining Step 5 with the lemma above gives the explicit finite-sample bound

$$\mathbb{E}[\text{Cost}] \;\leq\; \frac{d}{\alpha}\left(|I_{C_0}|\log C_0 - (N'_{C_0} - i_0 + 1)\left[\ln\sigma + \tfrac{1}{2}\ln\left(2\,\ln\left(\tfrac{N'_{C_0}}{k + N'_{C_0}\varepsilon_{N'_{C_0}}}\right)\right)\right]\right) + c_1\frac{d}{\alpha}\,\ln\ln\left(\frac{N'_{C_0}}{k + N'_{C_0}\varepsilon_{N'_{C_0}}}\right),$$

for a universal constant $c_1 > 0$. Moreover, since the per-candidate work is at most $d$, the unconditional expected cost satisfies

$$\mathbb{E}[\text{Cost}] \;\leq\; \mathbb{E}[\text{Cost} \mid \mathcal{E}_\delta]\,(1 - \delta) + \delta\,N'd \;\leq\; \frac{1}{1-\delta}\,\mathbb{E}[\text{Cost} \mid \mathcal{E}_\delta] + \delta\,N'd,$$

which yields the same bound up to an additive $\delta N'd$ and a multiplicative $1/(1 - \delta)$ factor.

**Comparison to naive cost**  The naive, brute-force method computes $N'$ full $d$-dimensional distances, with total cost at most $N'd$. Comparing with the bound above shows a reduction factor that scales as $\alpha$ (up to the slowly varying and logarithmic terms), on the same high-probability event $\mathcal{E}_\delta$.

**On the role of $\alpha > 1$**  The parameter $\alpha$ controls the rate of exponential energy decay, $e^{-\alpha m/d}$. If $\alpha \leq 1$, energy decays too slowly (e.g., at halfway, $m = d/2$, the remaining energy is at least $e^{-0.5}$), leading to weak bounds and limited pruning. Effective transforms concentrate energy early, which in practice corresponds to $\alpha$ comfortably greater than 1. The high-probability analysis simply replaces the expected-margin terms by their concentrated counterparts and leaves this qualitative conclusion unchanged.

ROBUSTNESS TO OUT-OF-DISTRIBUTION QUERIES

In practice, the query vector $\mathbf{q}$ and database vectors $\{\mathbf{x}_i\}$ may have different energy compaction properties under the learned transform $T$. Let $\alpha_q$ denote the energy compaction parameter for the query and $\alpha_x$ for the database vectors, such that:

$$R_{\mathbf{q}}^{(m,d)} \approx \|\mathbf{q}\|^2 e^{-\alpha_q m/d} \tag{7}$$

$$R_{\mathbf{x}_i}^{(m,d)} \approx \|\mathbf{x}_i\|^2 e^{-\alpha_x m/d} \tag{8}$$

**Theorem 8** (Effective energy compaction with asymmetric parameters). *When the query and database vectors have different compaction rates, the effective energy compaction parameter for the lower bound error becomes:*

$$\alpha_{\text{eff}} = \frac{\alpha_q + \alpha_x}{2}$$

*leading to an expected complexity of:*

$$\mathbb{E}[Cost] \sim \frac{C \cdot N'd}{\alpha_{\text{eff}}} \sim \frac{2C \cdot N'd}{\alpha_q + \alpha_x}$$

*for some constant $C > 0$ depending on the problem parameters.*

*Proof.* Starting from the same Cauchy-Schwarz derivation as in Step 2, the lower bound error is:

$$\|\mathbf{q} - \mathbf{x}_j\|^2 - \text{LB}^{(m)}(\mathbf{q}, \mathbf{x}_j) \leq 4\sqrt{R_{\mathbf{q}}^{(m,d)} R_{\mathbf{x}_j}^{(m,d)}}$$

With asymmetric energy compaction parameters, the tail energies become:

$$R_{\mathbf{q}}^{(m,d)} \leq \|\mathbf{q}\|^2 e^{-\alpha_q m/d} \leq R^2 e^{-\alpha_q m/d} \tag{9}$$

$$R_{\mathbf{x}_j}^{(m,d)} \leq \|\mathbf{x}_j\|^2 e^{-\alpha_x m/d} \leq R^2 e^{-\alpha_x m/d} \tag{10}$$

Substituting into the Cauchy-Schwarz bound:

$$4\sqrt{R_{\mathbf{q}}^{(m,d)} R_{\mathbf{x}_j}^{(m,d)}} \leq 4R^2 \sqrt{e^{-\alpha_q m/d} \cdot e^{-\alpha_x m/d}} = 4R^2 e^{-(\alpha_q+\alpha_x)m/(2d)}$$

The effective energy compaction parameter is therefore $\alpha_{\text{eff}} = (\alpha_q + \alpha_x)/2$, and the rest of the analysis follows identically to the symmetric case, yielding the stated complexity. □

**Graceful degradation for OOD queries** This result has important practical implications. Even when the query is completely out-of-distribution and exhibits no energy compaction ($\alpha_q = 0$), the algorithm still achieves a speedup factor of $\alpha_x/2$ compared to the naive approach:

$$\mathbb{E}[\text{Cost}] \sim \frac{2C \cdot N'd}{\alpha_x}$$

This demonstrates that Panorama provides robust performance even for challenging queries that don't conform to the learned transform's assumptions, maintaining substantial computational savings as long as the database vectors are well-compacted.

FINAL COMPLEXITY RESULT AND COMPARISON WITH NAIVE ALGORITHM

The naive brute-force algorithm computes the full $d$-dimensional distance for each of the $N'$ candidates, yielding cost $\text{Cost}_{\text{naive}} = N' \cdot d$.

**Theorem 9** (Main Complexity Result - Proof of Theorem 2). *Let $\phi = \frac{\mathbb{E}[\rho]}{d}$ be the average fraction of dimensions processed per candidate as defined in Section 3. Under assumptions A1–A4, the expected computational cost is:*

$$\mathbb{E}[Cost] = \phi \cdot d \cdot N' \sim \frac{C \cdot N'd}{\alpha}$$

*where $C$ can be made arbitrarily close to 1 through appropriate scaling.*

*Proof.* From Steps 1–6, the expected cost is approximately:

$$\mathbb{E}[\text{Cost}] \approx \frac{d}{\alpha}\left(|I_{C_0}| \log C_0 - (N'_{C_0} - i_0 + 1)\left[\ln \sigma + \frac{1}{2}\ln\left(2\ln\left(\frac{N'_{C_0}}{k + N'_{C_0}\varepsilon_{N'_{C_0}}}\right)\right)\right]\right)$$

For large $N'$, we have $\frac{|I_{C_0}|}{N'} \to 1$ and $\frac{N'_{C_0} - i_0 + 1}{N'} \to 1$, giving:

$$\phi = \frac{\mathbb{E}[\text{Cost}]}{N' \cdot d} \approx \frac{1}{\alpha}\left(\log C_0 - \ln \sigma - \zeta\right)$$

where $\zeta := \frac{1}{2}\ln\left(2\ln\left(\frac{N'_{C_0}}{k + N'_{C_0}\varepsilon_{N'_{C_0}}}\right)\right)$.

**Scaling to achieve $C = 1$.** Scale all vectors by $\beta > 0$: this transforms $R \to \beta R$ and $\sigma \to \beta\sigma$. The expression becomes:

$$\phi \approx \frac{1}{\alpha}\left(\log(\beta^2 C_0) - \ln(\beta\sigma) - \zeta\right) = \frac{1}{\alpha}\left(\log C_0 + 2\log\beta - \ln \sigma - \ln\beta - \zeta\right)$$

$$= \frac{1}{\alpha}\left(\log C_0 + \log\beta - \ln \sigma - \zeta\right)$$

By choosing $\beta = e^{\ln \sigma - \log C_0 + \zeta}$, we get $\log C_0 + \log\beta = \ln \sigma + \zeta$, making the leading coefficient exactly 1. Therefore $\phi \sim 1/\alpha$ and $\mathbb{E}[\text{Cost}] \sim N'd/\alpha$.

Note that $\zeta$ depends on the problem size $N'$, the number of nearest neighbors $k$, and the concentration parameter $\varepsilon_{N'_{C_0}}$.

This gives the asymptotic speedup: $\text{Cost}_{\text{naive}}/\mathbb{E}[\text{Cost}_{\text{Panorama}}] \sim \alpha$.

$\square$

## B    EXPERIMENTAL SETUP

### B.1    HARDWARE AND SOFTWARE

All experiments are conducted on Amazon EC2 `m6i.metal` instances equipped with Intel Xeon Platinum 8375C CPUs (2.90GHz), 512GB DDR4 RAM, running Ubuntu 24.04.3 LTS, and compiled with GCC 13.3.0. In line with the official ANN Benchmarks (Aumüller et al., 2020), all experiments are executed on a single core with hyper-threading (SMT) disabled.

Our code is publicly available at `https://anonymous.4open.science/r/panorama-iclr`.

### B.2    DATA COLLECTION

We benchmark each index using recall, the primary metric of the ANN Benchmarks (Aumüller et al., 2020). For each configuration, we run 100 queries sampled from a held-out test set, repeated 5 times. On HNSW, Annoy, and MRPT, build times for SIFT100M would commonly exceed 60 minutes. Since we conducted hundreds of experiments per index, we felt it necessary to use SIFT10M for these indexes to enable reasonable build times. All the other indexes were benchmarked using SIFT100M.

**IVFFlat and IVFPQ.**    Both methods expose two parameters: (i) $n_{\text{list}}$, the number of coarse clusters (256–2048 for most datasets, and 10 for CIFAR-10/FashionMNIST, matching their class counts), and (ii) $n_{\text{probe}}$, the number of clusters searched (1 up to $n_{\text{list}}$, sweeping over 6–10 values, primarily powers of two). IVFPQ additionally requires: (i) $M$, the number of subquantizers (factors of $d$ between $d/4$ and $d$), and (ii) $n_{\text{bits}}$, the codebook size per subquantizer (fixed to 8 (Jégou et al., 2011), yielding $M$ bytes per vector).

**HNSW.**    We set $M = 16$ neighbors per node (Malkov & Yashunin, 2020), $ef_{\text{construction}} = 40$ for index creation (Douze et al., 2024), and vary $ef_{\text{search}}$ from 1 to 2048 in powers of two.

**Annoy.**    We fix the forest size to $n_{\text{trees}} = 100$ (Bernhardsson, 2013) and vary $search\_k$ over 5–7 values between 1 and 400,000.

**MRPT.**    MRPT supports autotuning via a target recall (Jääsaari et al., 2019b), which we vary over 12 values from 0.0 to 1.0.

### B.3    DATA PROCESSING

For each index, we sweep its parameters and compute the Pareto frontier of QPS–recall pairs. To denoise, we traverse points from high to low recall: starting with the first point, we retain only those whose QPS exceeds the previously selected point by a factor of 1.2–1.5. This yields smooth QPS–recall curves. To obtain speedup–recall plots, we align the QPS–recall curves of the baseline and PANORAMA-augmented versions of an index, sample 5 evenly spaced recall values along their intersection, and compute the QPS ratios. The resulting pairs are interpolated using PCHIP.

### B.4    MODEL TRAINING

We trained Cayley using the Adam optimizer with a learning rate of 0.001, running for up to 100 epochs with early stopping (patience of 10). Training typically converged well before the maximum epoch limit, and we applied a learning-rate decay schedule to stabilize optimization and avoid overshooting near convergence. This setup ensured that PCA-Cayley achieved stable orthogonality while maintaining efficient convergence across datasets. The training was performed on the same CPU-only machine described in B, using 30% of the data for training and an additional 10% as a validation

set to ensure generalization. Since our transforms are not training-heavy, training usually finished in under 20 minutes for each dataset, except for SIFT (due to its large size) and Large/CIFAR-10 (3072-dimensional), where the training step took about 1 hour.

## B.5 Accounting for Transformation Time

PANORAMA applies an orthogonal transform to each query via a $1 \times d$ by $d \times d$ matrix multiplication. We measure this amortized cost by batching 100 queries per dataset and averaging runtimes using NumPy (Harris et al., 2020) on the CPUs of our EC2 instances. Table 3 reports the estimated maximum per-query transformation time share across datasets and index types. These values are accounted for in post-processing for every experiment we ran.

|        | Ada       | CIFAR-10  | FashionMNIST | GIST      | Large     | SIFT      |
|--------|-----------|-----------|--------------|-----------|-----------|-----------|
| Annoy  | 3.0e-04%  | 5.2e-03%  | 7.0e-03%     | 2.2e-04%  | 4.5e-04%  | 1.1e-04%  |
| HNSW   | 1.4e-02%  | 5.5e-02%  | 3.3e-02%     | 4.7e-03%  | 1.9e-02%  | 2.5e-04%  |
| IVFFlat| 1.1e-03%  | 1.5e-02%  | 1.8e-02%     | 8.1e-04%  | 1.3e-03%  | 1.7e-05%  |
| IVFPQ  | 2.7e-03%  | 8.4e-03%  | 7.0e-03%     | 6.7e-04%  | 2.2e-03%  | 3.3e-05%  |
| MRPT   | 1.7e-03%  | 1.7e-02%  | 1.1e-02%     | 5.5e-04%  | 3.0e-03%  | 5.9e-05%  |
| L2Flat | 7.0e-04%  | 5.6e-02%  | 1.3e-02%     | 7.0e-04%  | 8.5e-04%  | 1.4e-06%  |

Table 3: Estimated maximum per-query transform time (% of query time) by index and dataset.

## C PANORAMA VARIANTS

| Variant       | $|B|$      | Use UB | Applicable Indexes |
|---------------|------------|--------|--------------------|
| Point-centric | 1          | No     | HNSW, Annoy, MRPT  |
| Batch-UB      | $B \gg 1$  | Yes    | IVFPQ              |
| Batch-noUB    | $B > 1$    | No     | L2Flat, IVFFlat    |

Table 4: Panorama execution variants, parameterized by batch size ($B$) and whether upper bounds (UBs) are maintained.

The generic Panorama algorithm (Algorithm 4) is flexible and admits three execution modes depending on two factors: the batch size $B$ and whether we maintain *upper bounds* (UBs) during iterative refinement. We highlight three important variants that cover the spectrum of practical use cases. In each case, we present the pseudocode along with a discussion of the design tradeoffs and a summary in Table 4

### C.1 Point-centric: Batchsize = 1, Use $\pi = 0$

As outlined in Alg. 2, candidates are processed individually, with heap updates only after exact distances are computed. Since exact values immediately overwrite looser bounds, maintaining UBs offers no benefit. This mode is best suited for non-contiguous indexes (e.g., HNSW, Annoy, MRPT), where the storage layout is not reorganized. Here, pruning is aggressive and immediate. A candidate can be discarded as soon as its lower bound exceeds the current global threshold $d_k$. The heap is updated frequently, but since we only track one candidate at a time, the overhead remains low.

### C.2 Batch-UB: Batchsize $\neq$ 1, Use $\pi = 1$

As described in Alg. 3, when we process candidates in large batches ($B1$), the situation changes. Frequent heap updates may seem expensive, however, maintaining upper bounds allows us to prune more aggressively: a candidate can be pushed into the heap early if its UB is already tighter than the current $d_k$, even before its exact distance is known. When batch sizes are large, the additional

---

**Algorithm 2** PANORAMA: Point Centric

1: **Input:** Query $\mathbf{q}$, candidate set $\mathcal{C} = \{\mathbf{x}_1, \ldots, \mathbf{x}_{N'}\}$, transform $T$, levels $m_1 < \cdots < m_L$, $k$, batch size $B$
2: **Precompute:** $T(\mathbf{q})$, $\|T(\mathbf{q})\|^2$, and tail energies $R_q^{(\ell, d)}$ for all $\ell$
3: **Initialize:** Global exact distance heap $H$ (size $k$), global threshold $d_k \leftarrow +\infty$, $p(\mathbf{q}, \mathbf{x}) \leftarrow 0^{(l,l)}$
4: Compute exact distances of first $k$ candidates, initialize $H$ and $d_k$
5: **for** each candidate $\mathbf{x} \in \mathcal{C}$ **do**                                                    ▷ Batch $\mathcal{B} = \{p\}$
6:     **for** $\ell = 1$ to $L$ **do**
7:         **if** $\mathrm{LB}^\ell(\mathbf{q}, \mathbf{x}) > d_k$ **then**                                     ▷ Update LB bound
8:             Mark $\mathbf{x}$ as pruned                                                ▷ If threshold exceeded, prune candidate
9:             **continue**
10:        Push $(\mathrm{LB}^L(\mathbf{q}, \mathbf{x}), \mathbf{x})$ to $H$ as exact entry                 ▷ $\mathrm{LB}^L(\mathbf{q}, \mathbf{x})$ is ED as $\ell = L$
11:        **if** $d < d_k$ **then**
12:            Update $d_k = k^{\mathrm{th}}$ distance in $H$; Crop $H$
13: **return** Candidates in $H$ (top $k$ with possible ties at $k^{\mathrm{th}}$ position)

---

pruning enabled by UBs outweighs the overhead of heap updates. This tighter pruning is particularly beneficial in high-throughput, highly-optimized settings such as IVFPQ, where PQ compresses vectors into shorter codes, allowing many candidates to be processed together.

---

**Algorithm 3** PANORAMA: Batched with UB

1: **Input:** Query $\mathbf{q}$, candidate set $\mathcal{C} = \{\mathbf{x}_1, \ldots, \mathbf{x}_{N'}\}$, transform $T$, levels $m_1 < \cdots < m_L$, $k$, batch size $B$
2: **Precompute:** $T(\mathbf{q})$, $\|T(\mathbf{q})\|^2$, and tail energies $R_q^{(\ell, d)}$ for all $\ell$
3: **Initialize:** Global exact distance heap $H$ (size $k$), global threshold $d_k \leftarrow +\infty$, $p(\mathbf{q}, \mathbf{x}) \leftarrow 0^{(l,l)}$
4: Compute exact distances of first $k$ candidates, initialize $H$ and $d_k$
5: **for** each batch $\mathcal{B} \subset \mathcal{C}$ of size $B$ **do**
6:     **for** $\ell = 1$ to $L$ **do**
7:         **for** each candidate $\mathbf{x} \in \mathcal{B}$ **do**
8:             **if** $\mathrm{LB}^\ell(\mathbf{q}, \mathbf{x}) > d_k$ **then**                                 ▷ Update LB bound
9:                 Mark $\mathbf{x}$ as pruned                                            ▷ If threshold exceeded, prune candidate
10:                **continue**
11:            Compute $\mathrm{UB}^\ell(\mathbf{q}, \mathbf{x})$                                              ▷ Compute upper bound
12:            **if** $\mathrm{UB}^\ell(\mathbf{q}, \mathbf{x}) < d_k$ **then**
13:                Push $(\mathrm{UB}^\ell(\mathbf{q}, \mathbf{x}), \mathbf{x})$ to $H$ as UB entry
14:                Update $d_k = k^{\mathrm{th}}$ distance in $H$; Crop $H$
15: **return** Candidates in $H$ (top $k$ with possible ties at $k^{\mathrm{th}}$ position)

---

## C.3  BATCH-NOUB: BATCHSIZE $\neq 1$, USE $\pi = 0$

Finally, when batch size is greater than one but we disable UBs, we obtain a different execution profile, as described in Alg 4 In this mode, each batch is processed level by level, and pruning is done only with lower bounds. Candidates that survive all levels are compared against the global $d_k$ using their final exact distance, but the heap is updated only once per batch rather than per candidate. This reduces UB maintenance overhead, at the expense of weaker pruning within the batch. For L2Flat and IVFFlat, batch sizes are modest and candidates are uncompressed. Here, the marginal pruning benefit from UBs is outweighed by the overhead of heap updates, making UB maintenance inefficient.

---

**Algorithm 4** PANORAMA: Batched without UB

1: **Input:** Query $\mathbf{q}$, candidate set $\mathcal{C} = \{\mathbf{x}_1, \ldots, \mathbf{x}_{N'}\}$, transform $T$, levels $m_1 < \cdots < m_L$, $k$, batch size $B$
2: **Precompute:** $T(\mathbf{q})$, $\|T(\mathbf{q})\|^2$, and tail energies $R_q^{(\ell, d)}$ for all $\ell$
3: **Initialize:** Global exact distance heap $H$ (size $k$), global threshold $d_k \leftarrow +\infty$, $p(\mathbf{q}, \mathbf{x}) \leftarrow 0^{(l,l)}$
4: Compute exact distances of first $k$ candidates, initialize $H$ and $d_k$
5: **for** each batch $\mathcal{B} \subset \mathcal{C}$ of size $B$ **do**
6:     **for** $\ell = 1$ to $L$ **do**
7:         **for** each candidate $\mathbf{x} \in \mathcal{B}$ **do**
8:             **if** $\mathrm{LB}^\ell(\mathbf{q}, \mathbf{x}) > d_k$ **then**                                 ▷ Update LB bound
9:                 Mark $\mathbf{x}$ as pruned                                            ▷ If threshold exceeded, prune candidate
10:                **continue**
11:        **for** each unpruned candidate $\mathbf{x} \in \mathcal{B}$ **do**
12:            Push $(\mathrm{LB}^L(\mathbf{q}, \mathbf{x}), \mathbf{x})$ to $H$ as exact entry                 ▷ $\mathrm{LB}^L(\mathbf{q}, \mathbf{x})$ is ED as $\ell = L$
13:            **if** $d < d_k$ **then**
14:                Update $d_k = k^{\mathrm{th}}$ distance in $H$; Crop $H$
15: **return** Candidates in $H$ (top $k$ with possible ties at $k^{\mathrm{th}}$ position)

---

This setting is not equivalent to the point-centric case above. Here, all candidates in a batch share the same pruning threshold for the duration of the batch, and the heap is only updated at the end. This is the design underlying IVFFlat: efficient to implement, and still benefiting from level-major layouts and SIMD optimizations.

**Systems Perspective.** As noted in Section 3, these three Panorama variants capture a spectrum of algorithmic and systems tradeoffs:

- **Point-centric** ($B = 1$, $\pi = 0$): Suited for graph-based or tree-based indexes (Annoy, MRPT, HNSW) where candidates arrive sequentially, pruning is critical, and system overhead is minor.

- **Batch-UB** ($B1$, $\pi = 1$): Ideal for highly optimized, quantization-based indexes (IVFPQ) where aggressive pruning offsets the cost of frequent heap updates.

- **Batch-noUB** ($B1$, $\pi = 1$): Matches flat or simpler batched indexes (IVFFlat), where streamlined execution and SIMD batching outweigh the benefit of UBs.

# D HNSW: NON-TRIVIAL ADDITION

---

**Algorithm 5** HNSW + PANORAMA at Layer 0

---

1: **Input:** Query $\mathbf{q}$, neighbors $k$, beam width $efSearch$, transform $T$
2: **Initialize:** Candidate heap $C$ (size $efSearch$, keyed by partial distance), result heap $W$ (size $k$, keyed by exact distance), visited set $\{ep\}$ (entry point)
3: Compute $ed \leftarrow \|T(\mathbf{q}) - ep\|_2$
4: Insert $(ed, ep)$ into $C$ and $W$
5: **while** $C$ not empty **do**
6:     $v \leftarrow C.\text{pop\_min}()$
7:     $\tau \leftarrow W.\text{max\_key}()$ if $|W| = k$ else $+\infty$
8:     **for** each neighbor $u$ of $v$ **do**
9:         **if** $u \notin visited$ **then**
10:             Add $u$ to visited
11:             $(lb, ub, pruned) \leftarrow$ PANORAMA$(\mathbf{q}, u, T, \tau)$
12:             Insert $\left(\left(\frac{lb+ub}{2}\right), u\right)$ into $C$; crop $C$
13:             **if** not pruned **then**
14:                 Insert $(lb, u)$ into $W$; crop $W$         $\triangleright lb = ub = ed$
15: **return** Top-$k$ nearest elements from $W$
16:
17: **procedure** PANORAMA$(\mathbf{q}, u, T, \tau)$
18:     **for** each level $\ell$ **do**
19:         $lb \leftarrow \text{LB}^\ell(T(\mathbf{q}), u)$
20:         $ub \leftarrow \text{UB}^\ell(T(\mathbf{q}), u)$
21:         **if** $lb > \tau$ **then**
22:             **return** $(lb, ub, true)$         $\triangleright$ Pruned
23:     **return** $(lb, ub, false)$

---

HNSW constructs a hierarchical proximity graph, where an edge $(v, u)$ indicates that the points $v$ and $u$ are close in the dataset. The graph is built using heuristics based on *navigability*, *hub domination*, and *small-world properties*, but importantly, these edges do not respect triangle inequality guarantees. As a result, a neighbor's neighbor may be closer to the query than the neighbor itself.

At query time, HNSW proceeds in two stages:

1. **Greedy descent on upper layers:** A skip-list–like hierarchy of layers allows the search to start from a suitable entry point that is close to the query. By descending greedily through upper layers, the algorithm localizes the query near a promising root in the base layer.

2. **Beam search on layer 0:** From this root, HNSW maintains a candidate beam ordered by proximity to the query. In each step, the closest element $v$ in the beam is popped, its neighbors $N(v)$ are examined, and their distances to the query are computed. Viable neighbors are inserted into the beam, while the global result heap $W$ keeps track of the best $k$ exact neighbors found so far.

**Integration Point.** The critical integration occurs in how distances to neighbors $u \in N(v)$ are computed. In vanilla HNSW, each neighbor's exact distance to the query is evaluated immediately upon consideration. With Panorama, distances are instead refined progressively. For each candidate

$v$ popped from the beam heap, and for each neighbor $u \in N(v)$, we invoke PANORAMA with the current $k$-th threshold $\tau$ from the global heap:

- If Panorama refines $u$ through the final level $L$ and $u$ survives pruning, its exact distance is obtained. In this case, $u$ is inserted into the global heap and reinserted into the beam with its exact distance as the key.
- If Panorama prunes $u$ earlier at some level $\ell < L$, its exact distance is never computed. Instead, $u$ remains in the beam with an approximate key $(\mathsf{LB}^\ell + \mathsf{UB}^\ell)/2$, serving as a surrogate estimate of its distance.

**Heuristics at play.** This modification introduces two complementary heuristics:

- **Best-first exploration:** The beam remains ordered, but now candidates may carry either exact distances or partial Panorama-based estimates.
- **Lazy exactness:** Exact distances are only computed when a candidate truly needs them (i.e., it survives pruning against the current top-$k$). Non-viable candidates are carried forward with coarse estimates, just sufficient for ordering the beam.

**Why this is beneficial.** This integration allows heterogeneous precision within the beam: some candidates are represented by exact distances, while others only by partial Panorama refinements. The global heap $W$ still guarantees correctness of the final $k$ neighbors (exact distances only), but the beam search avoids unnecessary exact computations on transient candidates. Thus, HNSW+Panorama reduces wasted distance evaluations while preserving the navigability benefits of HNSW's graph structure.

# E  IVFPQ: IMPLEMENTATION DETAILS

We now describe how we integrated PANORAMA into Faiss's IVFPQ index. Our integration required careful handling of two performance-critical aspects: (i) maintaining SIMD efficiency during distance computations when pruning disrupts data contiguity, and (ii) choosing the most suitable scanning strategy depending on how aggressively candidates are pruned. We address these challenges through a buffering mechanism and a set of adaptive scan modes, detailed below.

**Buffering.** For IVFPQ, the batch size $B$ corresponds to the size of the coarse cluster currently being scanned. As pruning across refinement levels progresses, a naive vectorized distance computation becomes inefficient: SIMD lanes remain underutilized because codes from pruned candidates leave gaps. To address this, we design a buffering mechanism that ensures full SIMD lane utilization. Specifically, we allocate a 16KB buffer once and reuse it throughout the search. This buffer stores only the PQ codes of candidates that survive pruning, compacted contiguously for efficient SIMD operations. Buffer maintenance proceeds as follows:

1. Maintain a byteset where `byteset[i]` indicates whether the $i$-th candidate in the batch survives. We also keep a list of indices of currently active candidates.
2. While unprocessed points remain in the batch and the buffer is not full, load 64 bytes from the byteset (`_mm512_loadu_si512`).
3. Load the corresponding 64 PQ codes.
4. Construct a bitmask from the byteset, and compress the loaded codes with `_mm512_maskz_compress_epi8` so that surviving candidates are packed contiguously.
5. Write the compacted codes into the buffer.

Once the buffer fills (or no codes remain), we compute distances by gathering precomputed entries from the IVFPQ lookup table (LUT), which stores distances between query subvectors and all $2^{n_{\text{bits}}}$ quantized centroids. Distance evaluation reduces to `_mm512_i32gather_ps` calls on the buffered codes, and pruning proceeds in a fully vectorized manner.

**Scan Modes.** Buffering is not always optimal. If no candidates are pruned, buffering is redundant, since the buffer merely replicates the raw PQ codes. To avoid unnecessary overhead, we introduce a `ScanMode :: Full`, which bypasses buffering entirely and directly processes raw codes.

Conversely, when only a small fraction of candidates survive pruning, buffer construction becomes inefficient: most time is wasted loading already-pruned codes. For this case, we define `ScanMode :: Sparse`, where we iterate directly over the indices of surviving candidates in a scalar fashion, compacting them into the buffer without scanning the full batch with SIMD loads.

# F ABLATION STUDIES

We conduct multiple ablation studies to analyze the effect of individual components of PANORAMA, providing a detailed breakdown of its behavior under diverse settings.

The base indexes we use expose several knobs that control the QPS–recall tradeoff. An ANNS query is defined by the dataset (with distribution of the metric), the number of samples $N$, and the intrinsic dimensionality $d$. Each query retrieves $k$ out of $N$ entries. In contrast, PANORAMA has a single end-to-end knob, the hyperparameter $\alpha$, which controls the degree of compaction.

## F.1 TRUNCATION VS. PANORAMA

Vector truncation (e.g., via PCA) is often used with the argument that it provides speedup while only marginally reducing recall. However, truncating all vectors inevitably reduces recall across the board. In contrast, PANORAMA adaptively stops evaluating dimensions based on pruning conditions, enabling speedup *without recall loss*. Figure 11 shows % dimensions pruned (x-axis), recall (left y-axis), and speedup on L2Flat (right y-axis). The black line shows PANORAMA's speedup. To achieve the same speedup as PANORAMA, PCA truncation only achieves a recall of 0.58.

Figure 11: Truncation vs. PANORAMA: recall and speedup tradeoff.

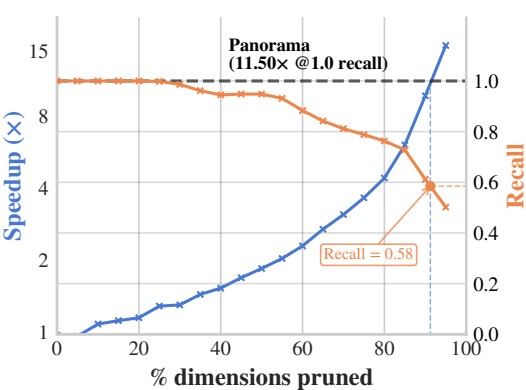

## F.2 ABLATION ON $N, d, k$

We do an ablation study on GIST1M using L2Flat to study the impact of the number of points, the dimension of each vector, and $k$ in the $k$NN query.

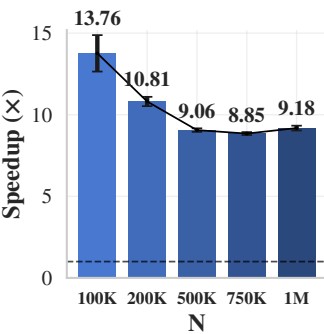

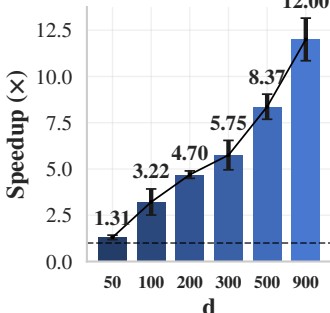

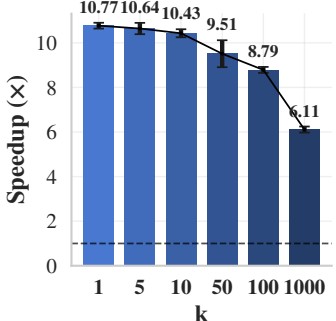

Figure 12: We study the effect of dataset size on GIST using L2Flat. In principle speedups should not depend on $N$ as we see for 500K - 1M, however nuances in selection of subset show higher speedups for 100K.

Figure 13: On GIST, we sample dimensions 10, 200, 300, 500, and 960, apply the Cayley transform, and measure speedup as $d$ varies.

Figure 14: We study scaling with $k$. We set $\max k = \sqrt{N}$, the largest value used in practice. Since the first $k$ elements require full distance computations, the overhead increases with $k$, reducing the relative speedup

### F.3 ABLATION ON PCA AND DCT

The above table compares PCA with Cayley transforms. It highlights the importance of having $\alpha$ (introduced in Section 5) as a tunable parameter. The following results show speedup on IVFPQ and clearly demonstrate how Cayley achieves superior speedups compared to PCA

| Dataset @recall | DCT ($\times$) | PCA ($\times$) | Cayley ($\times$) |
|---|---|---|---|
| Ada @98.0% | 1.675 | 4.196 | **4.954** |
| CIFAR-10 @92.5% | N/A | 2.426 | **3.564** |
| FashionMNIST @98.0% | 1.199 | 2.635 | **4.487** |
| GIST1M @98.0% | 2.135 | 6.033 | **15.781** |
| Large @98.0% | 5.818 | 12.506 | **15.105** |
| SIFT100M @92.5% | 0.821 | 3.842 | **4.586** |

Table 5: DCT vs. PCA vs. Cayley (IVFPQ).

or DCT methods. Despite the fact that DCT provides immense energy compaction on image datasets (CIFAR-10 and FashionMNIST), the transformed data ultimately loses enough recall on IVFPQ to render the speedups due to compaction underwhelming.

### F.4 ABLATION ON TRAINING SIZE

To directly assess how training data volume affects transformation quality, we trained on the GIST1M dataset at 16 levels and measured the percentage of dimensions scanned. Remarkably, even when using only 1% of the training data, the resulting Cayley matrices were nearly indistinguishable from those learned with the full dataset, demonstrating the robustness of the transforms to limited training data.

| Training samples (% of total) | % dimensions scanned |
|---|---|
| 10,000 (1.00%) | 6.87% |
| 100,000 (10.00%) | 6.83% |
| 1,000,000 (100.00%) | 6.82% |

Table 6: Impact of training data volume on transformation quality.

This strong empirical robustness of PANORAMA aligns with our theoretical characterization in Theorem 3, which establishes that Panorama is expected to maintain high compaction efficacy even for out-of-distribution (OOD) queries.

### F.5 ABLATION ON NUMBER OF THREADS

As noted in Appendix B, all experiments are executed in a single-threaded fashion. However, increasing´ the number of threads yields higher speedups, as pruning decreases memory pressure. An experiment with IVFFlat on GIST1M reveals the following speedup values:

| Number of threads | Speedup ($\times$) |
|---|---|
| 1 | 7.36$\times$ |
| 4 | 7.27$\times$ |
| 8 | 7.37$\times$ |
| 16 | 7.85$\times$ |
| 32 | 7.99$\times$ |

Table 7: Speedup scaling with number of threads on IVFFlat (GIST1M).

### F.6 N LEVELS ABLATION

Figure 15 highlights two key observations for GIST on IVFPQ under our framework:

**Impact of the number of levels.** Increasing the number of levels generally improves speedups up to about 32–64 levels, beyond which gains plateau and can even decline. This degradation arises from the overhead of frequent pruning decisions: with more levels, each candidate requires more branch evaluations, leading to increasingly irregular control flow and reduced performance.

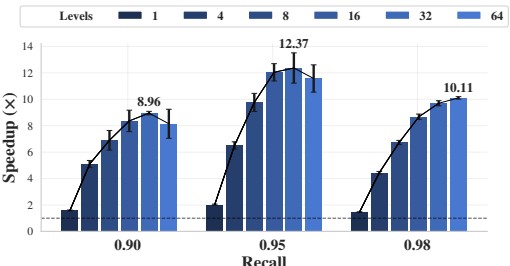

Figure 15: Speedups vs. number of levels.

**Cache efficiency from LUT re-use.** Panorama's level-wise computation scheme naturally reuses segments of the lookup table (LUT) across multiple queries, mitigating cache thrashing. Even in isolation, this design yields a $1.5 - 2\times$ speedup over standard IVFPQ in Faiss. This underscores that future system layouts should be designed with Panorama-style execution in mind, as they inherently align with modern cache and SIMD architectures.

### F.7 LB ABLATION

We evaluate the tightness of our lower bound using the ratio $\frac{\mathsf{LB}^\ell}{||x-q||}$, which measures how quickly the lower bound approaches the true distance. As shown in Figure 16, PANORAMA 's learned transform yields a rapidly increasing ratio, reaching close to the final distance within the first 20–30% of dimensions. This confirms that the pruning effectiveness comes from tighter distance bounds—which aligns with our Theorem 2 that connects compaction (Figure 6) with tightness of bounds (Figure 16).

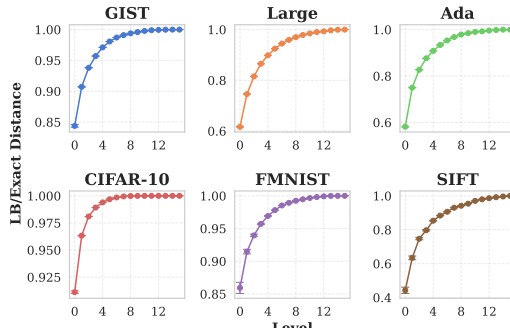

Figure 16: $\frac{\mathsf{LB}^\ell}{||x-q||}$ vs. levels.

## F.8   REAL VS. EXPECTED SPEEDUP

We compare the speedup predicted by our pruning model against the measured end-to-end speedup, validating both the analysis and the practical efficiency of our system. The *expected speedup* is a semi-empirical estimate: it takes the observed fraction $o$ of features processed and combines it with the measured fraction $p$ of time spent in verification. Formally,

$$s_{\exp} \;=\; \frac{1}{(1-p) + p \cdot o}.$$

When verification dominates ($p = 1$), this reduces to $s_{\exp} = 1/o$, while if verification is negligible ($p = 0$), no speedup is possible regardless of pruning. The *actual speedup* is measured as the ratio of PANORAMA 's end-to-end query throughput over the baseline, restricted to recall above 80%. Figure 17 shows that $s_{\exp}$ and the measured values closely track each other, confirming that our system implementation realizes the gains predicted by pruning, though this comparison should not be confused with our theoretical results.

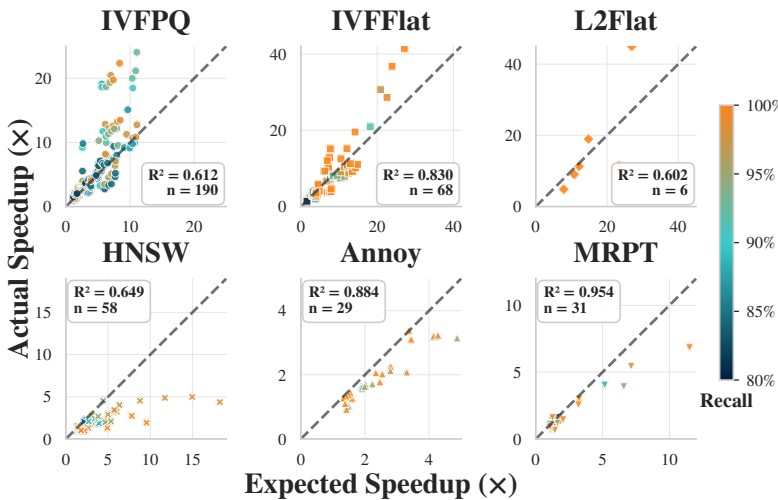

Figure 17: Comparison of measured and predicted speedup across datasets.

1) **Implementation gains.**   For IVFPQ—and to a lesser extent IVFFlat and L2Flat—measured speedups exceed theoretical predictions. This stems from reduced LUT and query-cache thrashing in our batched, cache-aware design, as explained in Section 6.

2) **Recall dependence.**   Higher recall generally comes from verifying a larger candidate set. This increases the amount of work done in the verification stage, leading to larger gains in performance (e.g., IVFPQ, HNSW).

3) **Contiguous indexes.**   Layouts such as IVFPQ and IVFFlat realize higher predicted speedups, since they scan more candidates and thus admit more pruning. Their cache-friendly structure allows us to match—and sometimes surpass due to (1)—the expected bounds.

4) **Non-contiguous indexes.** Graph- and tree-based methods (e.g., HNSW, Annoy, MRPT) saturate around 5–6× actual speedup across our datasets, despite higher theoretical potential. Here, cache misses dominate, limiting achievable gains in practice and underscoring Amdahl's law. Moreover, in Annoy and MRPT specifically, less time is spent in the verification phase overall.

### F.9 QPS VS. RECALL SUMMARY

Finally, Figure 18 summarizes the overall QPS vs. Recall tradeoffs across datasets and indexes.

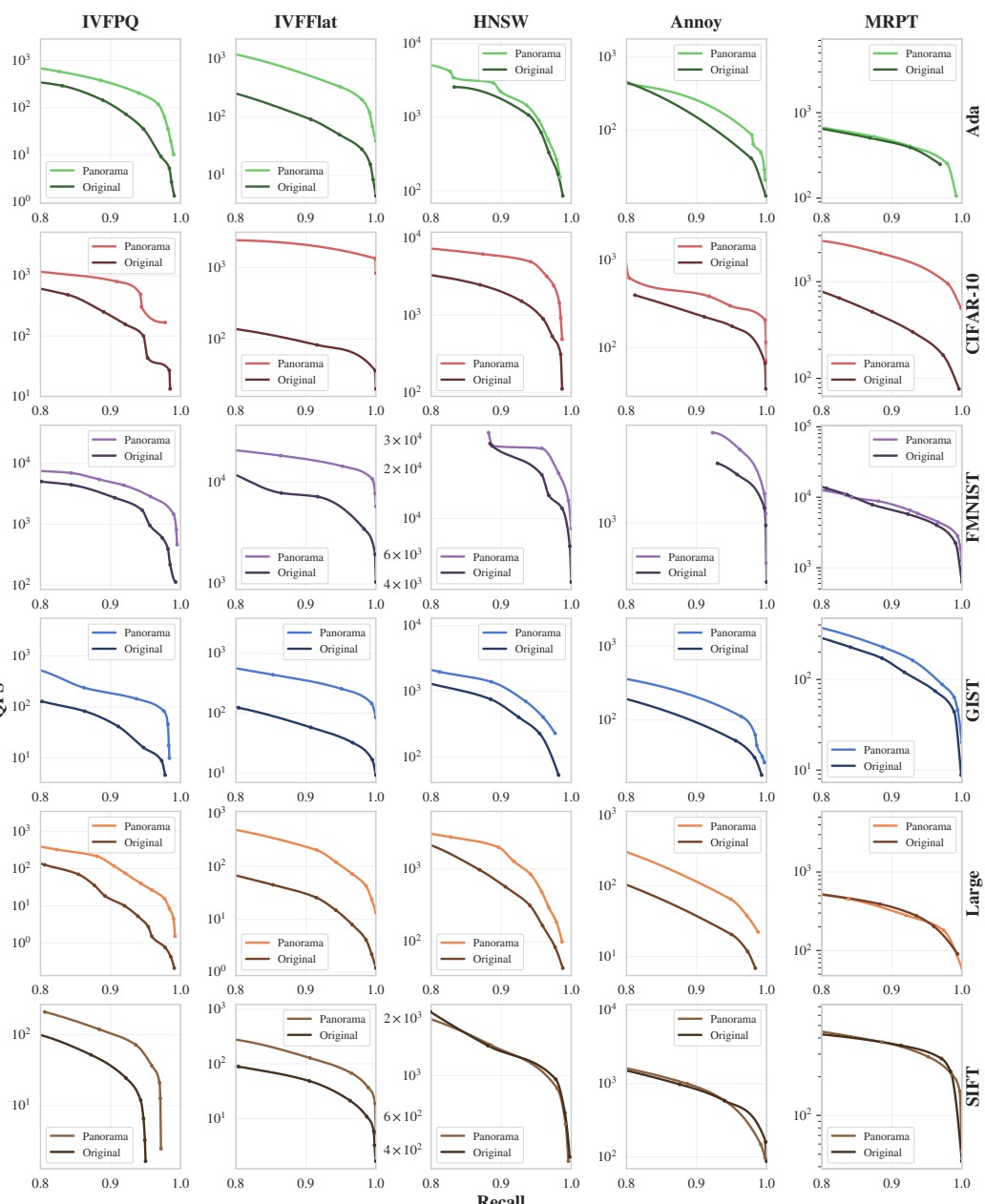

Figure 18: QPS vs. Recall: base index vs. PANORAMA+index across datasets.

QPS vs. recall plots are generated for every combination of index (PANORAMA and original) and dataset using the method outlined in Appendix B. These graphs are used to generate the Speedup vs. recall curves in Figure 8.

**LLM Usage Statement** We used an LLM to assist in polishing the manuscript at the paragraph level, including tasks such as re-organizing sentences and summarizing related work. All LLM-generated content was carefully proofread and verified by the authors for grammatical and semantic correctness before inclusion in the manuscript.

