# OpenReview forum: "Panorama: Fast-Track Nearest Neighbors"
_ICLR.cc/2026/Conference — Submitted to ICLR 2026_

### Official Review · Reviewer_6PQd · 2025-10-23

**Soundness:** 1
**Presentation:** 2
**Contribution:** 2
**Rating:** 2
**Confidence:** 5

**Summary:**

The paper proposes PANORAMA, a framework to accelerate the refinement phase in Approximate Nearest Neighbor Search (ANNS). The key idea is to use a learned orthogonal transformation that concentrates the norm of the data in the first few dimensions, allowing for early candidate pruning via tight lower bounds (LB) on the L2 distance. The authors integrate PANORAMA into existing ANNS systems (IVFPQ, HNSW, MRPT, and Annoy), demonstrating empirical speedups of 2-30× while maintaining recall.

The method builds upon the idea that a well-chosen orthogonal transform T can compact signal energy, thereby yielding tighter cumulative distance bounds (via Cauchy–Schwarz inequalities). A Cayley parameterization is used to learn T, and partial distances are computed incrementally to prune candidates when the lower bound exceeds the top-k distance threshold.

**Strengths:**

- The motivation is relevant: reducing the verification cost in ANNS is practically important.

- The integration into multiple ANNS backends is technically solid and shows strong engineering effort.

- Experimental results are extensive, and speedup claims are clearly reported.

**Weaknesses:**

**W1. Incorrect runtime accounting for the learned transformation**

The learned transform T is applied both to database vectors and queries (Section 4), but in experiments, the query time does not include the transformation cost—as visible from the released code (I checked simple_benchmark.py and see transformed queries are stored separately from original queries) and described pipelines (eq (1)). Since T(q) must be computed for every query, omitting this step underestimates query latency. For large d, the transformation cost (a dense matrix–vector multiply) can dominate the supposed savings from partial distance pruning.

**W2. Theoretical–empirical mismatch in the use of the lower bound**

The theoretical lower bound relies on decomposing ∥q−x∥ = ∥T(q)∥ + ∥T(x)∥ − 2 ⟨T(q),T(x)⟩ (Eq. 1). However, even if T compacts the energy of
x and q individually, it does not guarantee that the energy of their difference ∥q−x∥ is similarly front-loaded. Thus, the learned transform may tighten bounds on norms of x, but not necessarily on distances, breaking the link between the “energy compaction” assumption (A1) and the pruning efficacy. The experiments show speedups mainly from implementation optimizations rather than LB tightness. Fig 6 seems to reflect the concentration of the norm of data points only, not the distance. I checked evaluate_all_transformed_datasets.py in the released code, which seems to confirm my findings.

**W3. Novelty is limited, and the lack of relevant competitors**

The idea of leveraging partial or bounded L2 distances has been explored in several recent works, including Gao & Long (2023) and Yang et al. (2025), both cited by the authors. Those methods also used partial distance estimation or orthogonal projections to accelerate refinement. The proposed contribution proposes a new learning transformation that potentially gives tight lower-bound formulations. Hence, the improvement is primarily an engineering optimization (cache layout, batching, SIMD) rather than a new algorithmic insight. Also, the competitors are ANNS solvers without the use of lower bounds. They do not include relevant competitors (e.g. Gao & Long (2023) or standard methods: ANNS with DCT/FFT transformation) that use LB for speeding up the verification.

**W4. Empirical gains not well-attributed**

It is unclear how much of the reported 2–30× speedup originates from the theoretical contribution (learned transformation and bound) versus from memory layout changes (level-major batching) or engineering effort. The lack of ablation on the cost of the learned transform further blurs this distinction.

**W5. Unrealistic distributional assumptions**

Assumption A3 in Section 3 states that the squared distances ∥q−x∥ follow a Gaussian distribution. This is not true for high-dimensional candidate sets where all candidates are close to q. In practice, kNN candidates are not independent random samples—they cluster around
q. This invalidates parts of the theoretical derivation and weakens claims of Theorem 2.

**Minor:**
- Notations need to be consistent in the whole paper, i.e. candidate size (N' in Problem 1, N in Theorem 2), C in Theorem 1 and Theorem 2 and 3; so(d) vs SO(d)
- Theorem 1 is trivial.
- I think Subsections 4.1 and 4.2 should be significantly improved. I could not see the link between the learning objective in (6) and 4.1 subsection.
- Text font in Algorithm is smaller than the main text.

**Questions:**

Please address the raised weaknesses above, and some further questions

**Q1. Query-time transformation overhead.**

PANORAMA applies the learned orthogonal transformation T(q) to each query (Appendix B.5). What is the average per-query overhead when this transformation is included in the full pipeline timing (not isolated)? How significant is this cost relative to the total query latency, especially on high-dimensional datasets such as GIST or CIFAR-10?

**Q2. Tightness of the lower bound.**

How tight is the proposed PANORAMA lower bound on **distance** compared to other dimension-wise partial-distance bounds such as those obtained via FFT or DCT transforms. Could you quantify the actual distance-ratio gap (i.e., LB/∥x−q∥) using the same number of reduced dimensions to better interpret the bound’s tightness beyond speedup metrics?

**Q3. Threading and cache sensitivity.**

Are all query experiments executed in single-threaded or multi-threaded mode? Since the proposed lower-bound (LB) pruning mechanism relies on early termination within distance accumulation, concurrent threads may interfere due to cache sharing. How sensitive is the observed speedup to the degree of threading and CPU cache hierarchy?

**Q4. Training cost and data scale.**

Could you specify the runtime to learn the transformation compared to the cost of building indexes? Appendix B.4 mentions training times under 20 minutes (≈ 1 hour for SIFT and Large/CIFAR-10), which seem to be too large compared to building indexes used by Faiss.

---

> ### Author Response · Authors · 2025-11-22
> **Response to Reviewer 6PQd - Part 1**
>
> Thank you, Reviewer 6PQd, for your detailed technical review.
>
> ### W1/Q1: Transformation time overhead
> >the query time does not include the transformation cost—as visible from the released code (I checked simple_benchmark.py and see transformed queries are stored separately from original queries) and described pipelines (eq (1)).
> >Q1. What is the average per-query overhead when this transformation is included in the full pipeline timing (not isolated)? How significant is this cost relative to the total query latency, especially on high-dimensional datasets such as GIST or CIFAR-10?
>
> We do include the query transformation latency in all end-to-end measurements, and also report it in Table 3 (Appendix B.5). The transformation cost is **negligible**, as a simple matrix-vector multiplication ($d^2$ multiplications per query), made efficient by modern SIMD instructions and amortized across candidates. As Table 3 (Appendix B.5) shows, it takes at most **0.05% of total query time** across all datasets and indexes. For example:
> - SIFT100M with `IVFFlat`: 0.0017% of query time, and
> - GIST1M with `HNSW`: 0.0047% of query time.
>
> We note a typo: the caption of Table 3 should be "transform time (% of query time)", not "transform time (seconds)". We have fixed this.

---

> ### Author Response · Authors · 2025-11-22
> **Response to Reviewer 6PQd - Part 2**
>
> ### W2.1: Link between energy compaction and bound tightness
> >the learned transform may tighten bounds on norms of x, but not necessarily on distances.
>
> A transform that tightens bounds on vector norms automatically tightens distance bounds: the compaction of the energy of individual vectors $\mathbf{q}$ and $\mathbf{x}$ leads to small tail energies, and hence tight distance lower bounds. If $|T(\mathbf{x})_i|$ and $|T(\mathbf{q})_i|$ are small, then $(T(\mathbf{x})_i - T(\mathbf{q})_i)^2$ is also expected to be small.
>
> In our revision, we now demonstrate this empirically (Figure 16). This result is discussed in further details in our response to W2.3 below.
>
> ### W2.2: Source of speedups
> >The experiments show speedups mainly from implementation optimizations rather than LB tightness.
>
> Several of our experiments document speedups due to LB tightness:
>
> * Panorama achieves speedups up to **6×** on `HNSW` and `Annoy`, which do not employ system optimizations: speedups are entirely due to the algorithmic gain of reduced distance computations.
> * The ablation study in Figure 15 shows an up to 8.96X speedup in GIST1M due to inceasing the number of levels (F.6 in Appendix).
> * The measured end-to-end speedup tracks an implementation-agnostic **predicted speedup**, computed from the pruning fraction, with a ratio near 1.0. Our system design thus simply facilitates the algorithmic gains due to the reduction of distance computations (Sec F.8, Figure 17).
> * The ablation study in Section 7.4 (Figure 8) shows pruning without transforms yields up to 2× speedups.
>
> >how much of the speedup originates from theoretical contribution vs memory layout.
>
> We conduct ablations (Figure 8) to determine the gains due to each component. Across indexes on GIST1M, pruning alone yields between **1.05× and 2.04×** speedups, while transforms contribute significantly larger gains ranging from **1.83× to 5.20×**. The combined effect (pruning × transform) produces total speedups between **2.26× and 9.18×**:
>
> | Index    | Pruning Speedup | Transform Speedup | Total Speedup |
> |----------|------------------|-------------------|----------------|
> | IVFPQ    | 2.040×           | 4.153×            | 8.473×         |
> | IVFFlat  | 1.765×           | 5.198×            | 9.176×         |
> | HNSW     | 1.050×           | 4.115×            | 4.322×         |
> | Annoy    | 1.093×           | 2.073×            | 2.265×         |
> | MRPT     | 1.253×           | 1.830×            | 2.292×         |
>
>
> ### W2.3/Q2 Energy compaction for pruning (Theorem 2)
> >Fig 6 seems to reflect the concentration of the norm of data points only.
> >Q2. [...] How tight is the proposed PANORAMA lower bound on distance compared to other dimension-wise partial-distance bounds such as those obtained via FFT or DCT transforms. Could you quantify the actual distance-ratio gap (i.e., LB/∥x−q∥) [...]?
>
> Figure 6 presents the energy concentration of individual vectors, which entails pruning efficacy: **Theorem 2** proves that the expected pruning fraction is **inversely proportional** to the energy compaction parameter $\alpha$, $\mathbb{E}[\text{Cost}] \sim C \cdot \frac{N d}{\alpha}$, hence tighter energy compaction in vector norms (i.e., higher $\alpha$) leads to more effective pruning. The energy compaction of norms (Figure 6) validates our premise, and the end-to-end speedups (Figures 7 and 8) validate its implications. We have added a figure showing distance compaction via the ratio $\frac{LB}{||x-q||}$, which exhibits an exponentially decaying pattern, confirming the theory.
>
> To make this effect explicit, we also report the *distance compaction ratio* $(\frac{LB}{|x-q|})$ across refinement levels for all datasets in a new ablation (Figure 16, App. F7), summarized below. The ratio increases rapidly with the level (e.g., 4, 8, 12), exhibiting the expected exponential convergence implied by the theory.
>
> Levels chosen correspond to approximately **25%**, **50%**, and **75%** of total dimensions (≈ level 4, 8, 12 in the plot).
>
> | **Dataset**  | **Level 4** (≈25%) | **Level 8** (≈50%) | **Level 12** (≈75%) |
> | ------------ | ------------------ | ------------------ | ------------------- |
> | **GIST**     | 0.957089           | 0.991078           | 0.999018            |
> | **Large**    | 0.865332           | 0.959483           | 0.993083            |
> | **Ada**      | 0.876467           | 0.968043           | 0.992652            |
> | **CIFAR-10** | 0.989213           | 0.999454           | 0.999996            |
> | **FMNIST**   | 0.957099           | 0.989339           | 0.998175            |
> | **SIFT**     | 0.797146           | 0.929874           | 0.979637            |
>
> **Interpretation.**
> Across all datasets, the lower bound rapidly approaches the exact distance—exceeding **0.9 by level 8** and reaching **>0.99 by level 12** for almost all datasets. This strong “distance compaction” behavior directly underpins Panorama’s pruning aggressiveness and aligns with the theoretical exponential decay.

---

> ### Author Response · Authors · 2025-11-22
> **Response to Reviewer 6PQd - Part 3**
>
> ### W3: On missing competitors
> > The idea of leveraging partial or bounded L2 distances has been explored in several recent works, including Gao & Long (2023) and Yang et al. (2025) [...]. The proposed contribution proposes a new learning transformation that potentially gives tight lower-bound formulations. Hence, the improvement is primarily an engineering optimization [...] rather than a new algorithmic insight. Also, the competitors are ANNS solvers without the use of lower bounds. They do not include relevant competitors (e.g. Gao & Long (2023) or standard methods: ANNS with DCT/FFT transformation) that use LB for speeding up the verification.
>
> We have excluded those baselines as they are fundamentally different in terms of approximation vs. correctness: they provide probabilistic estimates relying on random projections ([Gao & Long, 2023](https://arxiv.org/abs/2303.09855)) or lack formal guarantees ([Yang et al., 2025](https://arxiv.org/abs/2404.16322)). We tried baselining against ADSampling [Gao & Long, 2023] but observed very unstable results. ADSampling relies on JL-based probabilistic guarantees, which assume data becomes sufficiently random after projection. This assumption fails on structured or OOD data, while Panorama’s guarantees remain valid. To demonstrate this distinction, we first construct a simple uniform dataset by sampling points in a hyperrectangle:
>
> | D   | N      | Recall (ADS) | Recall (Panorama) | Dims Scanned (ADS) | Dims Scanned (Panorama) |
> | --- | ------ | ------------ | ----------------- | ------------- | ------------------ |
> | 128 | 10000  | 1.000        | 1.000             | 99%         | 64%              |
> | 128 | 50000  | 1.000        | 1.000             | 93%         | 54%              |
> | 128 | 100000 | 1.000        | 1.000             | 92%         | 50%              |
> | 256 | 10000  | 1.000        | 1.000             | 98%         | 65%              |
> | 256 | 50000  | 1.000        | 1.000             | 76%         | 62%              |
> | 256 | 100000 | 1.000        | 1.000             | 96%         | 59%              |
> | 512 | 10000  | 1.000        | 1.000             | 99%         | 74%              |
> | 512 | 50000  | 1.000        | 1.000             | 89%         | 70%              |
> | 512 | 100000 | 1.000        | 1.000             | 97%         | 66%              |
>
> As the above table shows, on the given hyperparameters on the paper ($\epsilon$=2.1) ADSampling preserves recall but provides close to no pruning (large dims scanned %).
> While reducing $\epsilon$ to 1.0 allowed pruning comparable to Panorama there was a significant drop in recall (1.0 $\rightarrow$ 0.95).
>
> | Method         | ε       | Avg Recall | Dims Scanned                |
> | -------------- | ------- | ---------- | -------------------------- |
> | **ADSampling** | **2.1** | 0.9589     | 50%                        |
> | **ADSampling** | **1.0** | 1.0000     | 93%                        |
> | **Panorama**   | —       | 1.0000     | 63%                        |
>
> Then, we create a synthetic example with skewed data to show that ADSampling is prone to losing substantial recall on structured data, while Panorama maintains perfect recall and specially capitalises on this structure:
>
> | D   | N      | Recall (ADS) | Recall (Panorama) | Dims Scanned (ADS) | Dims Scanned (Panorama) |
> | --- | ------ | ------------ | ----------------- | ------------- | ------------------ |
> | 128 | 10000  | 0.880        | 1.000             | 74.8%         | 11.6%              |
> | 128 | 50000  | 0.860        | 1.000             | 58.4%         | 7.6%              |
> | 128 | 100000 | 0.870        | 1.000             | 51.8%         | 7.0%             |
> | 256 | 10000  | 0.620        | 1.000             | 72.6%         | 11.6%              |
> | 256 | 50000  | 0.630        | 1.000             | 56.5%         | 7.6%              |
> | 256 | 100000 | 0.630        | 1.000             | 54.4%         | 7.0%              |
> | 512 | 10000  | 0.330        | 1.000             | 71.6%         | 12.2%              |
> | 512 | 50000  | 0.320        | 1.000             | 58.4%         | 7.8%              |
> | 512 | 100000 | 0.310        | 1.000             | 54.0%         | 7.0%              |
>
>
> Our results highlight that ADSampling's reliability degrades on non-uniform or structured data, which are common in real-world applications, while Panorama maintains perfect recall with superior pruning efficiency. Panorama is, to our knowledge, the first ANN method to combine **learned loss-based transforms with lossless bound-based refinement**, guaranteeing no degradation in recall.
>
> FFT outputs complex-valued vectors, making it incompatible with L2-based pruning, while DCT is tailored for local spatial coherence (e.g., JPEG compression). Because **L2 is permutation-invariant while DCT is not**, arbitrary vector rotations eliminate its relevance outside image data.
>
> Lastly, as outlined in W2.2, our systems optimizations simply enable these theoretical speedups.

---

> > ### Author Response · Authors · 2025-11-22
> > **Response to Reviewer 6PQd - Part 4**
> >
> > #### W4.1: Algorithmic vs. systems contributions
> > >It is unclear how much of the reported 2–30× speedup originates from the theoretical contribution (learned transformation and bound) versus from memory layout changes (level-major batching) or engineering effort.
> >
> > Addressed in W2.2 above.
> >
> > #### W4.2/Q4: Cost of the learned transform
> > >The lack of ablation on the cost of the learned transform further blurs this distinction.
> > >Q4. Training cost and data scale: Could you specify the runtime to learn the transformation compared to the cost of building indexes? Appendix B.4 mentions training times under 20 minutes (≈ 1 hour for SIFT and Large/CIFAR-10), which seem to be too large compared to building indexes used by Faiss.
> >
> > The learned transform incurs a one-time offline training cost and a negligible per-query inference cost. Applying the transform $T$ to a query $\mathbf{q}$ requires a single dense matrix–vector multiplication. As discussed in W1, this overhead is **at most 0.05% of total query time**. The one-off training phase on the full GIST1M dataset took about **20 minutes** (Appendix B.4). Besides, the learned transform works even with tiny training sets as discussed in W2 of Reviewer HxV7:
> >
> > | Training Samples on GIST1M | % Dimensions Scanned |
> > |---------------------------:|----------------------:|
> > | 10,000 (1.0%)              | 6.87%                |
> > | 100,000 (10%)              | 6.83%                |
> > | 1,000,000 (100%)           | 6.82%                |
> >
> > Training on **just 1%** of the data, which takes **under 5 seconds**, achieves almost identical pruning effectiveness as training on the entire dataset. Training is thus a **minor one-off investment** with **long-term payoff**.
> >
> > ### W5: On Distributional Assumptions
> > > Assumption A3 in Section 3 states that the squared distances ∥q−x∥ follow a Gaussian distribution. This is not true for high-dimensional candidate sets where all candidates are close to q. In practice, kNN candidates are not independent random samples—they cluster around q. This invalidates parts of the theoretical derivation and weakens claims of Theorem 2.
> >
> > The Gaussian assumption (A3) is a standard tool used for analytical tractability in high-dimensional spaces, and Panorama's robustness does not critically depend on it.
> >
> > 1. **The assumption is not critical**
> >    We use the Gaussian model to infer the pruning margin's behavior, yet the final complexity result (**Theorem 2**) is remarkably robust. The speedup is primarily dictated by the energy compaction parameter $\alpha$, while the terms from the Gaussian assumption ($\mu$ and $\sigma$) are absorbed into a leading constant $C$. As we show in Appendix A, "Scaling to achieve $C = 1$", constant $C$ can be normalized away, leaving the $\alpha$-fold speedup as the dominant outcome. Even for distributions skewed toward the query (e.g., a beta distribution), the core of our algorithm holds: a tighter initial threshold $d_k$ from closer candidates enables aggressive pruning of other candidates.
> >
> > 2. **Validity in high dimensions**
> >    In high-dimensional spaces, the phenomenon known as *concentration of measure* causes distances between points to become more alike. The distance distribution thus concentrates around its mean, rendering a Gaussian (bell-shaped) approximation more appropriate for a large candidate set.
> >
> > 3. **Empirical validation overrides modeling simplifications:**
> >    Ultimately, the measure of a theoretical model lies in its predictive power. As **Table 2** shows, the pruning fractions our model predicts line up strikingly well with observed values across six datasets. This close match confirms that our assumptions capture the dynamics at play and lead to accurate real-world predictions.
> >
> > In summary, the Gaussian assumption is a reasonable modeling choice, while its details do not alter the final $\alpha$-fold speedup conclusion, which is validated by our experiments.

---

> ### Author Response · Authors · 2025-11-22
> **Response to Reviewer 6PQd - Part 5**
>
> ## Minor + Q&A
>
> >Notations need to be consistent in the whole paper, i.e. candidate size (N' in Problem 1, N in Theorem 2), C in Theorem 1 and Theorem 2 and 3; so(d) vs SO(d)
>
> - **N vs N'and C in Theorem 1, 2, 3**: We have made the change in the updated PDF.
> - **𝔰𝔬(d) vs SO(d)**: These are intentionally different - 𝔰𝔬(d) is the Lie algebra (tangent space) while SO(d) is the orthogonal group: 𝔰𝔬(d) is the tangent space to SO(d) at the identity.
>
> >Subsections 4.1 and 4.2 should be significantly improved. I could not see the link between the learning objective in (6) and 4.1 subsection.
>
> We have updated our manuscript based on your suggestion. Section 4.1 (now 5.1) shows how to maintain orthogonality (hence correctness of distance calculations) as a constraint, and Section 4.2 (now 5.2) shows how to achieve energy compaction as an objective. Equation 6 is the loss function for training the architecture in Section 4.1 (now 5.1), which relates to the objective of Section 4.2 (now 5.2).
>
> >Q3. Threading and cache sensitivity: Are all query experiments executed in single-threaded or multi-threaded mode? Since the proposed lower-bound (LB) pruning mechanism relies on early termination within distance accumulation, concurrent threads may interfere due to cache sharing. How sensitive is the observed speedup to the degree of threading and CPU cache hierarchy?
>
> All experiments are executed in a single-threaded fashion (as noted in Section B.1). Increasing the number of threads yields higher speedups, as pruning decreases memory pressure. An experiment with `IVFFlat` on GIST1M reveals the following speedup values:
>
> | Number of threads | Speedup (×) |
> |------------------:|------------:|
> | 1                 | 7.36×       |
> | 4                 | 7.27×       |
> | 8                 | 7.37×       |
> | 16                | 7.85×       |
> | 32                | 7.99×       |
>
> We have added this experiment and discussion to Appendix F.5 of our revised manuscript. Thank you for the thorough and constructive review.

---

> > ### Comment · Reviewer_6PQd · 2025-11-25
> >
> > Thanks a lot for the detailed feedback.
> >
> > There are still a couple of important points that I am struggling to reconcile with my understanding.
> >
> > __The transformation overhead on the query phase__
> >
> > You state that the transformation overhead in the query phase is negligible, but I am not sure how this matches the numbers.
> >
> > Since the transformation requires a matrix–vector multiplication, we can model this as roughly
> > $d$ distance computations. On GIST1M with HNSW ($n=10^6$, $d \approx 1000$), this overhead is reported as only 0.0047% of the query time. Even if I assume that a matrix–vector multiplication corresponds to about $d/10$ distance computations, i.e., roughly 100 distance computations, I still do not see how this can amount to only 0.0047% of HNSW’s query time. That fraction would imply that HNSW performs on the order of $> n$ distance computations per query, which seems implausible.
> >
> > Could you clarify how this percentage was obtained, or what assumptions underlie this estimate?
> >
> > __Link between energy compaction and bound tightness__
> >
> > I also disagree with the argument relating energy compaction to tighter distance bounds.
> >
> > A transform that tightens bounds on vector norms does not necessarily tighten bounds on pairwise distances. Consider the case where $x_1, x_2$ capture most of the energy of $x$, but $q_d, q_{d-1}$ capture most of the energy of
> > $q$. In such a situation, even if the transform compacts the energy of the data vectors into a few leading dimensions, the query’s energy may lie mostly in the trailing dimensions, so any distance-based bound that assumes aligned energy concentration may fail to tighten.
> >
> > This issue becomes even more critical for out-of-distribution (OOD) queries, where the energy of $q$ can be concentrated in essentially arbitrary dimensions. In that case, I do not see how energy compaction on the database side alone can guarantee tighter distance bounds.
> >
> > Could you please elaborate on how your argument handles such asymmetric or OOD scenarios?

---

> > > ### Author Response · Authors · 2025-11-26
> > > **Transformation overhead**
> > >
> > > Thank you for your engaging response.
> > >
> > > >Since the transformation requires a matrix–vector multiplication, we can model this as roughly $d$ distance computations. ... I still do not see how this can amount to only 0.0047% of HNSW’s query time.
> > >
> > > Indeed, there was an error in the original table: we inadvertently divided the transform time for a single query by the search time for a batch of 100 queries. After correcting this error, all reported ratios increase by a factor of 100. Even by the corrected ratios, however, the transformation accounts for only 0.47% of the end-to-end query time.
> > >
> > > The ratio remains small because a dense $d \times d$ matrix–vector multiplication is substantially cheaper than to $d \times d$ distance computations in practice. A Euclidean distance requires subtractions and squarings, whose constant factors exceed those of simple multiply–add operations. By contrast, the matrix–vector multiplication is implemented almost entirely using fused multiply–add (FMA) instructions, which modern CPUs execute extremely efficiently.
> > >
> > > All updated measurements were produced using the script at `transforms/transform_time.py`.

---

> > > ### Author Response · Authors · 2025-11-26
> > > **Link between energy compaction and bound tightness**
> > >
> > > Thank you for raising this cardinal question:
> > >
> > > >Consider the case where $x_1$, $x_2$ capture most of the energy of $x$, but $q_d$, $q_{d-1}$ capture most of the energy of $q$. ... I do not see how energy compaction on the database side alone can guarantee tighter distance bounds.
> > >
> > > The key insight is that pruning depends on the *product* of residual energies ($\sqrt{R_q \cdot R_x}$). Even if the query is not compacted, the rapid decay of the database vector's energy drives the error term to zero. Besides, if energies are concentrated in disjoint dimensions, the vectors are effectively orthogonal ($\langle q, x \rangle \approx 0$), making the true (squared) distance simply the sum of norms, to which the bound rapidly converges.
> > >
> > > **Numerical Example:**
> > > Consider a 4-dimensional space with a pruning threshold $d_k = 0.5$ established by a true nearest neighbor.
> > > *   **Query ($q$):** $[0, 0, 1, 1]$ (OOD, energy in tail). $\|q\|^2=2$.
> > > *   **Candidate ($x$):** $[1, 1, 0, 0]$ (Compacted, energy in head). $\|x\|^2=2$.
> > > *   **True Distance:** $\|q - x\|^2 = 4$ (Candidate should be pruned).
> > >
> > > **Step 1: Process Dimensions 1 & 2 (First half)**
> > > *   **Partial distance**: the algorithm reads the first dimensions, $x=[1, 1]$, corresponding to $q=[0, 0]$; the partial distance accumulates a contribution from $x$'s norm, $1^2 + 1^2 = 2$.
> > > *   **Residuals**: $R_x = 0$ and $R_q = 2$.
> > > *   **Error bound**: proportional to $\sqrt{R_x \cdot R_q} = \sqrt{0 \cdot 2} = 0$.
> > > *   **Result:** By dimension 2, the bound becomes tight: $LB = 2 + 2 - 0 = 4$.
> > > *   Since $LB = 4 > d_k = 0.5$, **$x$ is successfully pruned**.
> > >
> > > This result aligns with **Theorem 8**, which states that the effective convergence rate is the average of the query and database compaction rates: $\alpha_{eff} = (\alpha_q + \alpha_x)/2$. Even if $\alpha_q \approx 0$, a high $\alpha_x$ (well-compacted database) ensures the error term dimishes quickly and leads to effective pruning.

---

> > > > ### Comment · Reviewer_6PQd · 2025-11-26
> > > >
> > > > Thank you for the clarification.
> > > >
> > > > I still do not understand why you use a batch of 100 queries in your experiments, given that your implementation runs on a single thread and the multi-threaded version does not exhibit any thread-scaling speed-up.
> > > >
> > > > Assuming your estimate of 0.5% query-time overhead is correct, then the algorithm may indeed become attractive when the number of candidates is sufficiently large—particularly for high-dimensional datasets where high recall requires examining many vectors. For example, on GIST, if I treat matrix–vector multiplication as equivalent to roughly d/10≈100 squared-L2 distance computations, then your transformation becomes negligible once the candidate set reaches 20K or more. In this regime the overhead is fully amortised.
> > > >
> > > > Regarding the connection between lower bounds and energy compaction: thanks again for your additional clarification. In my earlier example I overlooked the role of the residual norms used in the lower bound. My current understanding is that you need Assumption A1 to hold for both x and q. If the coordinates of q are closer to uniform, or if they follow a mixture-of-Gaussians structure, then the lower bound will likely be very loose because the residual norms will not approximate the true residual distances well.
> > > >
> > > > ------------
> > > >
> > > > After the rebuttal, I have updated my score to 4. However, I still cannot fully support acceptance. My main concern is that I cannot tell whether the reported improvements stem mainly from the theoretical contribution (the learned transformation and the lower bound), or from the engineering choices (e.g., memory-layout changes such as level-major batching). The absence of comparisons against prior work that proposed similar ideas (Gao & Long, 2023; Yang et al., 2025) further makes it difficult for me to reconcile the experimental gains with my understanding of the underlying methods.

---

> ### Author Response · Authors · 2025-11-26
> **The reported improvements stem mainly from the theoretical contribution**
>
> Thank you for the prompt response.
>
> >I still do not understand why you use a batch of 100 queries in your experiments.
>
> We run realistic batches of 100 queries, in accordance with what Faiss and the official ANN Benchmarks support. This significantly reduces per-query overhead, such as the cost of computing the linear transform, which is also computed in a batch.
>
> >the multi-threaded version does not exhibit any thread-scaling speed-up.
>
> As we show in the updated Appendix F.5, we maintain a speedup of 7-8x on IVFFlat as the number of threads increases from 1 to 32.
>
> >you need Assumption A1 to hold for both $x$ and $q$.
>
> Theorem 3 establishes that it is not necessary that $q$ follows A1; $x$ follows A1 thanks to the learned transform trained on a loss function to that end. As discussed, we have verified experimentally that bound-based pruning still works with OOD queries.
>
> >After the rebuttal, I have updated my score to 4.
>
> Thank you for this reconsideration.
>
> >I cannot tell whether the reported improvements stem mainly from the theoretical contribution (the learned transformation and the lower bound), or from the engineering choices (e.g., memory-layout changes such as level-major batching).
>
> As we discussed, the measured end-to-end speedup tracks the implementation-agnostic semi-empirical **predicted speedup**, which we compute from the observed pruning fraction. Their ratio stays near 1.0 (Sec F.8, Figure 17), strongly suggesting that the reported improvements stem mainly from the theoretical contribution; our system co-design simply preserves the gains from pruning distance computations due to our algorithm and transform.
>
> >The absence of comparisons against prior work that proposed similar ideas (Gao & Long, 2023; Yang et al., 2025) further makes it difficult for me to reconcile the experimental gains with my understanding of the underlying methods.
>
> We already provided results against ADSampling [Gao & Long, 2023] on a simple uniform dataset constructed by sampling points in a hyperrectangle:
>
> | D   | N      | Recall (ADS) | Recall (Panorama) | Dims Scanned (ADS) | Dims Scanned (Panorama) |
> | --- | ------ | ------------ | ----------------- | ------------- | ------------------ |
> | 128 | 10000  | 1.000        | 1.000             | 99%         | 64%              |
> | 128 | 50000  | 1.000        | 1.000             | 93%         | 54%              |
> | 128 | 100000 | 1.000        | 1.000             | 92%         | 50%              |
> | 256 | 10000  | 1.000        | 1.000             | 98%         | 65%              |
> | 256 | 50000  | 1.000        | 1.000             | 76%         | 62%              |
> | 256 | 100000 | 1.000        | 1.000             | 96%         | 59%              |
> | 512 | 10000  | 1.000        | 1.000             | 99%         | 74%              |
> | 512 | 50000  | 1.000        | 1.000             | 89%         | 70%              |
> | 512 | 100000 | 1.000        | 1.000             | 97%         | 66%              |
>
> As the above table shows, on the hyperparameters given in the paper ($\epsilon$=2.1) ADSampling preserves recall but provides close to no pruning. Reducing $\epsilon$ to 1.0 allowed pruning comparable to Panorama, yet there was a significant drop in recall.
>
> | Method         | ε       | Avg Recall | Avg Pruning                |
> | -------------- | ------- | ---------- | -------------------------- |
> | **ADSampling** | **2.1** | 0.9589     | 50%                        |
> | **ADSampling** | **1.0** | 1.0000     | 93%                        |
> | **Panorama**   | —       | 1.0000     | 63%                        |
>
> We also created a synthetic skewed example, in which ADSampling's random projection assigns most of the energy of the true nearest neighbour at their rear end. ADSampling then prunes away the true nearest neighbor, based on the low-energy initial coefficients, and loses substantial recall, while Panorama maintains perfect recall with superior pruning efficiency:
>
> | D   | N      | Recall (ADS) | Recall (Panorama) | Dims Scanned (ADS) | Dims Scanned (Panorama) |
> | --- | ------ | ------------ | ----------------- | ------------- | ------------------|
> | 128 | 10000  | 0.880        | 1.000             | 74.8%         | 11.6%             |
> | 128 | 50000  | 0.860        | 1.000             | 58.4%         | 7.6%              |
> | 128 | 100000 | 0.870        | 1.000             | 51.8%         | 7.0%              |
> | 256 | 10000  | 0.620        | 1.000             | 72.6%         | 11.6%             |
> | 256 | 50000  | 0.630        | 1.000             | 56.5%         | 7.6%              |
> | 256 | 100000 | 0.630        | 1.000             | 54.4%         | 7.0%              |
> | 512 | 10000  | 0.330        | 1.000             | 71.6%         | 12.2%             |
> | 512 | 50000  | 0.320        | 1.000             | 58.4%         | 7.8%              |
> | 512 | 100000 | 0.310        | 1.000             | 54.0%         | 7.0%              |

---

### Official Review · Reviewer_HxV7 · 2025-10-28

**Soundness:** 4
**Presentation:** 3
**Contribution:** 4
**Rating:** 8
**Confidence:** 2

**Summary:**

ANNS is a highly practical research direction with extensive applications in industry. The authors of this paper propose a data-adaptive transformation method that compresses feature discriminablity (energy) into the leading dimensions without losing structural relationships. Combined with pruning, this approach reduces the computational cost of feature similarity calculations. The method can be jointly applied with all four existing classic ANNS algorithms to further enhance retrieval efficiency.

**Strengths:**

1）The research problem has very significant practical application value.
2）The theoretical proofs are comprehensive.
3）The proposed method is compatible with all four existing classic approaches, further improving retrieval efficiency.
4）It not only introduces an algorithm but also provides optimization solutions at the system level.

**Weaknesses:**

1）Lack of large-scale experiments. The largest dataset used in the paper is only 1 million in scale, which is relatively small for practical industrial applications. For instance, the datasets in previous NeurIPS competitions have already reached the billion scale.
2）Generalizability of the data-driven transformation matrix. The quality of this transformation matrix is likely highly dependent on the training dataset, raising concerns about its generalization capability.
3）Substantial reconstruction overhead when integrated with methods like IVFFlat. The underlying reconstruction effort required for integration is non-trivial.

**Questions:**

Please refer to the weaknesses section.

---

> ### Author Response · Authors · 2025-11-22
> **Response to Reviewer HxV7**
>
> We sincerely thank you, Reviewer HxV7, for your positive assessment and valuable feedback.
>
> ### W1: Dataset scale
> >The largest dataset used in the paper is only 1 million in scale.
>
> **We use 100-million data sets**: we use SIFT100M (Table 1); Panorama delivers substantial gains of 2–30× on this 100M scale (Figure 7). For HNSW, Annoy, and MRPT, we only use SIFT10M, as their build times on SIFT100M exceed one hour. Panorama does not change the behavior of the underlying index in the filtering phase, hence **preserves the scalability of methods that already scale to billion-vector datasets** and accelerates their refinement phase.
>
>
> ### W2: Transformation quality
> >The quality of the transformation matrix is likely highly dependent on the training dataset.
>
> To directly assess how training data volume affects transformation quality, we trained on the GIST1M dataset at 16 levels and measured the percentage of dimensions scanned. Remarkably, even when using only 1% of the training data, the resulting Cayley matrices were nearly indistinguishable from those learned with the full dataset, demonstrating the robustness of the transforms to limited training data. This experiment and discussion has been added to Appendix F.4 (Line 1530) of the revised manuscript.
>
> | Training samples (% of total) | % dimensions scanned |
> |-------------------------------|-----------------------|
> | 10,000 (1.00%)                | 6.87%                 |
> | 100,000 (10.00%)              | 6.83%                 |
> | 1,000,000 (100.00%)           | 6.82%                 |
>
> This strong empirical robustness of Panorama aligns with our theoretical characterization in Theorem 3, which establishes that Panorama is expected to maintain high compaction efficacy even for out-of-distribution (OOD) queries. We also validate this robustness empirically in Fig. 9. Using synthetic OOD queries of varying difficulty on IVFFlat with the GIST1M dataset, PANORAMA consistently delivers substantial speedups—7.89× for the most challenging cases (RC = 1), 8.86× for moderately difficult queries (RC = 3), and 11.51× for in-distribution queries (Figure 9). Moreover, the learned transforms generalize well across diverse data modalities (images, text embeddings, scene descriptors) without any dataset-specific tuning.
>
> ### W3: Reconstruction overhead with `IVFFlat`
> >Substantial reconstruction overhead when integrated with methods like IVFFlat.
>
> If reconstruction refers to integrating Panorama, we note that **we have integrated Panorama within Faiss' IVFFlat** and achieve 2-40× speedups (Figure 7) (Figure 17). If reconstruction refers to the inverse transform, we note that this is a straightforward operation unaffected by IVFFlat's layout. We will be happy to clarify any other aspect.

---

> ### Author Response · Authors · 2025-11-28
> **Eagerly waiting for feedback on revisions made**
>
> Dear Reviewer HxV7,
>
> We have provided detailed explanations and additional experiments to address your concerns. We have also uploaded a revised manuscript with all new additions clearly highlighted in blue font. Since the discussion phase will close soon, we are keen to get your reaction to the changes made.
>
> Regards,
>
> Authors

---

### Official Review · Reviewer_6394 · 2025-11-01

**Soundness:** 2
**Presentation:** 2
**Contribution:** 3
**Rating:** 4
**Confidence:** 3

**Summary:**

This paper studies the problem of optimizing the *refinement* phrase of approximate near neighbor (ANN) search, which the authors define as identifying the top $k$ elements from an initial match set of $|\mathcal{C}| > k$ items. The authors introduce a novel framework called Panorama to solve the refinement problem by leveraging orthogonal transforms. In particular, the authors introduce data-adaptive learned orthogonal transforms based on the Cayley transform over the Stiefel manifold that aims to concentrate vector distances in the initial dimensions and thus reduce the computational complexity of distance calculations during the refinement phase. In addition to theoretical guarantees, the authors also implement their approach by carefully considering the memory layout of the underlying ANN algorithm and report substantial speedups on the order of 2-30x for the refinement phase.

**Strengths:**

1. The proposed method introduced in the paper of energy-based learned orthogonal transforms is very creative and a novel application within this particular problem domain.

2. The authors work in combining rigorous theoretical guarantees with high-performance implementations that consider low-level memory layout details is a strong effort in bridging theory and systems work, which is a rare combination in the retrieval literature.

3. The ideas in the paper have the potential to inspire further work along this direction.

4. The authors proposed approach is very general and can be applied to virtually any ANN algorithm regardless of its inner workings.

**Weaknesses:**

In my opinion, the biggest weakness in the current version of the paper is an insufficient discussion of related work on this topic. The notion of refinement in retrieval and similarity search is very well studied and goes by various names such as "reranking" and "approximate distance computations." I think it would be very helpful if the authors included a dedicated related work section that discussed prior approaches to refinement. Moreover, I believe it is critical for the authors to compare against previously published techniques in their experimental evaluation and thereby consider more rigorous baselines than naive refinement. In particular, prior works that I think are very relevant to this paper and should be discussed include: (1) [Finger](https://arxiv.org/pdf/2206.11408), (2) [Probabilistic Kernel Function for Angle Testing](https://arxiv.org/pdf/2505.20274), and (3) [A Bi-metric Framework for Fast Similarity Search](https://arxiv.org/pdf/2406.02891) (plus the broader literature on reranking techniques). I believe that addressing and experimentally evaluating against this prior literature is critical for positioning this new work appropriately.

In addition, I think it would be very helpful if the authors considered additional large-scale benchmark datasets at the 100M or 1B vector scale, such as those from Big ANN Benchmarks.

**Questions:**

1. Can the authors provide a more thorough discussion of prior published work in the refinement literature, including perhaps the papers listed above (if they are in fact relevant)? I think it is critical to include this discussion in the paper in a standalone section.

2. Can the authors also provide an experimental comparison with previously published refinement algorithms that go beyond naive refinement? Additional experiments on large-scale datasets, such as those from Big ANN benchmarks, might also be very helpful in supporting the claims made in the paper.

---

> ### Author Response · Authors · 2025-11-22
> **Response to Reviewer 6394**
>
> Thank you Reviewer 6394 for your constructive feedback and suggestions for strengthening our work.
>
> ### W1: Related work on probabilistic refinement and reranking
> >consider more rigorous baselines than naive refinement: (1) Finger, (2) Probabilistic Kernel Function for Angle Testing, and (3) A Bi-metric Framework for Fast Similarity Search (plus the broader literature on reranking techniques).
>
> We have now expanded the related work section with the mentioned works.
>
> **Probabilistic refinement** techniques such as [Finger](https://arxiv.org/pdf/2206.11408), [Probabilistic Kernel Methods](https://arxiv.org/pdf/2505.20274), and the [Bi-metric Framework](https://arxiv.org/pdf/2406.02891) employ probabilistic bounds or cheap but inaccurate proxy metrics, which allow for **recall degradation** and are confined to **graph-based indexes** (i.e., HNSW/DiskANN-style graphs). Contrariwise, Panorama provides **exact bounds**, hence does not compromise recall, while being applicable to the refinement phase of **any** filtering-and-refinement algorithm. We clarified this distinction in the related work section.
>
> **Reranking** is the name used for the refinement step applied on a large ($k \gg 1000$) coarse intermediate candidate pool retrieved via codeword bucketization in [IVFPQFastScan](https://faiss.ai/cpp_api/struct/structfaiss_1_1IndexIVFPQFastScan.html) and bitwise encoding in [RaBitQ](https://arxiv.org/pdf/2405.12497). Panorama is applicable to the **refinement (reranking) stage** of this workflow too. We show an example using our Panorama-enhanced Faiss `IndexFlatL2`, leaving the base index unchanged (vanilla `IVFPQFastScan`) yielding **up to 2× speedup**:
>
> #### **Table: Reranking Performance on GIST1M (IVF256, PQ60×4fs, nprobe=16, nlevels=8, k=10)**
>
> | k_factor | k_base (= k × k_factor) | Speedup   | Recall ($r$) |
> | -------- | ----------------------- | --------- | ---------- |
> | 1        | 10                      | 1.00×     | 0.04       |
> | 8        | 80                      | 1.01×     | 0.16       |
> | 64       | 640                     | **1.25×** | 0.47       |
> | 256      | 2560                    | **1.70×** | 0.70       |
> | 1024     | 10240                   | **2.15×** | 0.82       |
>
> As the coarse candidate pool grows (larger ($k_\text{base})$), the refinement cost dominates. Reranking thus offers an additional task to which Panorama applies and provides benefits.
>
> ### W2: Dataset scale
> >consider additional large-scale benchmark datasets at the 100M or 1B vector scale.
>
> **Our experiments already use such datasets**: we use SIFT100M (Table 1); Panorama delivers substantial gains of 2–30× on this 100M scale (Figure 7). For HNSW, Annoy, and MRPT, we only use SIFT10M, as their build times on SIFT100M exceed one hour. Panorama does not change the behavior of the underlying index in the filtering phase, hence **preserves the scalability of methods that already scale to billion-vector datasets** and accelerates their refinement phase.

---

> ### Author Response · Authors · 2025-11-28
> **Eagerly waiting for feedback on revisions made**
>
> Dear Reviewer 6394,
>
> We have provided detailed explanations and additional experiments to address your concerns. We have also uploaded a revised manuscript with all new additions clearly highlighted in blue font. Since the discussion phase will close soon, we are keen to get your reaction to the changes made.
>
> Regards,
>
> Authors

---

### Official Review · Reviewer_YBgs · 2025-11-03

**Soundness:** 3
**Presentation:** 3
**Contribution:** 2
**Rating:** 2
**Confidence:** 2

**Summary:**

This work studied a method to improve the efficiency of approximate nearest neighbor search with no performance loss. The approximate nearest neighbor search algorithms are fast, but this work combines with naive kNN L2Flat (Douze et al., 2024) which performs
a brute-force kNN search over the entire dataset. The computation cost of kNN grows fast when the feature dimension of dataset goes large. The authors propose a multiple level, multiple batches method to index the dataset, the level 1 (leaf level) has M dimension which equals the number of features of the dataset.

According to the Algorithm 1, that the computation is linear with parameters. However, it is not sure if the memory cost is same as Problem 1 2^|D|, $|D|$ is the number of samples. For retrieval question, the challenge also comes from large number of dataset pool. If memory cost is $2^|D|$, it is not desirable.

The lower bound and upper bound as shown in Equation 3 and 4 are not informative enough. In a sense, the bound do not contribute a lot to the theoretical guarantee.

Also the main contribution is the speedup without recall loss. If the number of retrieved samples is too big, the retrieved results are not useful. The number of returned samples is not specified in the experimental results. If authors could help solve the concerns, that would be helpful.

**Strengths:**

The problem is interesting and with practical use. The introduction introduce the most recent works that this work is most related. That is helpful to understand the main contribution of the work.

There are multiple dimensions of experiments to validate the speedup contribution of the work.

**Weaknesses:**

The lower bound and upper bound are loose, they do not show contributions to the theoretical guarantee.

The memory cost is 2^|D|, $|D|$ is the number of samples which is huge in retrieval question. The multiple-level indexing method as shown in Figure 3 seems computational expensive to me, since the retrieval requires inner product computation from tree root to leaf. It does not make use of any correlation between samples in batches, batches.

The recall loss claim does not specify the number of returned samples, if the recall loss has the trade of the number of retrieved samples, the retrieval results are not informative enough.

**Questions:**

1. According to the Algorithm 1, that the computation is linear with parameters. However, it is not sure if the memory cost is same as Problem 1 2^|D|, $|D|$ is the number of samples.

2. How does the lower bound and upper bound as shown in Equation 3 and 4 contribute to the theoretical guarantee?

3. What is the average number of retrieval for each query for different datasets in the experiments?

---

> ### Author Response · Authors · 2025-11-22
> **Response to Reviewer YBgs**
>
> Thank you, Reviewer YBgs, for your thoughtful review. We provide detailed clarifications below.
>
> ### W1.1 Contribution of Equation 3,4
> >The lower bound and upper bound in Equation 3 and 4 are not informative enough.
> >How do the bounds in Equation 3 and 4 contribute to the theoretical guarantee?
>
> These Cauchy–Schwarz bounds lead to Theorem 2 in combination with the energy compaction property of the learned orthogonal transform, establishing a theoretical guarantee on the expected computational cost to process a candidate set of size N. This connection is clarified in Lines 936-940 (appendix) of our manscript.
>
> Specifically:
> * Equations 3 and 4 bound the inner product $\left| \sum_{j=m+1}^{d} T(q)\_j T(x)\_j \right|$ by the square root of the product of residual energies, $\sqrt{ R\_{T(\mathbf{q})}^{(\ell,d)} R\_{T(\mathbf{x})}^{(\ell,d)}}$ (Lines 170-173).
> * Thereby, refinement computations benefit from the exponential decay of those energies due to the orthogonal transform $T$, as $R_{T( \mathbf{q} )}^{(\ell,d)} \approx \|\mathbf{q}\|^2 e^{-\alpha \ell/d}$ and $R\_{T( \mathbf{x} )}^{(\ell,d)} \approx \|\mathbf{x}\|^2 e^{-\alpha \ell/d}$.
> * Consequently, $\left| \sum_{j=m+1}^{d} T(q)_j T(x)_j \right| \leq \|\mathbf{q}\| \|\mathbf{x}\| e^{-\alpha \ell/d}$.
>
> ### W1.2 Tightness of bounds
> >The lower bound and upper bound are loose.
>
> The Cauchy–Schwarz inequality provides the **tightest possible worst-case bound for inner products** without any structural assumptions and with computations no more expensive than the inner-product itself (see https://en.wikipedia.org/wiki/Cauchy%E2%80%93Schwarz_inequality#Properties. *Callebaut's Inequality*, which is tighter, requires computing partial powers.
>
> To further study the tightness of the bounds and their impact, in Appendix F.7 (Line 1585), we have an empirical study on the variation of LB/exact_distance and UB/exact_distance against dimensions. We show it rapidly approaches a tightness of 1 within 20%-30% of the dimensions.
>
>
> ### W2.1: Memory cost
> >not sure if the memory cost is $2^{|D|}$
>
> Panorama's memory footprint is **not** $2^{|D|}$. As detailed in Section 6.3, Panorama's memory overhead is $O(nL)$, where $n$ is the number of points in the dataset, and $L$ is the number of levels (typically 8-64). The term $2^{|D|}$ in Section 3 refers to **all candidate subsets**. Each level requires an additional single 4-byte float per data point, making the overhead very small, as reflected in the examples on GIST1M (Section 6.3):
> - **IVFPQ** with \(M = 480\), \(nbits = 8\), \(L = 8\): only **7.5%** additional storage, and
> - **IVFFlat** with (M = 8): only **0.94%** additional storage.
>
> ### W2.2: Multi-level indexing
> >The multiple-level indexing method as shown in Figure 3 seems computationally expensive to me, since the retrieval requires inner product computation from tree root to leaf.
>
> PANORAMA **does not** perform root-to-leaf traversal. Figure 3 shows a flat memory layout (**not an index**) that organizes features in batched, level-major order for SIMD efficiency. Distance (and inner-product) computation proceeds via sequential scans over this contiguous layout; no tree navigation occurs; the apparent "leaves" in the layout are the linearly accessed feature values.
>
> ### W3: Recall loss
> > The recall loss claim does not specify the number of returned samples.
>
> Panorama returns the same set of $k$ samples as standard refinement, hence **the same recall** as the base index. The Panorama IVFPQ version, which applies product quantization on transformed data, yields different recall from the standard version because product quantization does not preserve norms. Nevertheless, the recall is again determined by the base index.
>
> ### Retrieval per query
> >What is the average number of retrieval for each query for different datasets in the experiments?
>
> As mentioned in Appendix B.2 and in the main manuscript (Lines 435-436), for each configuration, we run 100 queries sampled from a held-out test set, repeated 5 times.

---

> ### Author Response · Authors · 2025-11-28
> **Eagerly waiting for feedback on revisions made**
>
> Dear Reviewer YBgs,
>
> We have provided detailed explanations and additional experiments to address your concerns. We have also uploaded a revised manuscript with all new additions clearly highlighted in blue font. Since the discussion phase will close soon, we are keen to get your reaction to the changes made.
>
> Regards,
>
> Authors

---

### Author Response · Authors · 2025-11-22
**Cover Letter**

We thank the reviewers for their thorough and thoughtful suggestions. A comprehensive response to the comments is presented below. **We have updated the main manuscript** to address these comments. The changes made in the manuscript are highlighted in **blue** font. The major changes are listed below:
 - **Additional Experiments**
   - Varying training set size for Cayley transforms (App. F.4)
   - Varying parallelism (number of threads) (App. F.5)
   - Observing progressive refinement of LB/exact_distance (App. F.7)
 - **Presentation**
   - A new Related Works section covering FINGER, Probabilistic Kernel Methods, Bi-metric Framework, and reranking.
   - Reworked Section 5 to have more clarity and explain connection to subsequent subsections.
   - Addressing a typo in App. B.5
   - Improved notation ($\Gamma$ for candidate set, standardized $N$ and $N'$, $\text{Cost}$ for Cost)

The revised manuscript remains within the 10 page limit of rebuttal.

---

### Comment · Area_Chair_RsmH · 2025-11-24
**Reviewer & Author Discussion**

Hi Reviewers,

Please kindly and actively participate in the review-author discussion if you haven't already, raise your further concerns so that the authors can explain more, and make your final decisions.

Best,
AC

---

### Meta-Review · Area_Chair_3p3D · 2026-01-04

**Summary:**

The concerns focus on novelty of the work, missing baselines, tightness of provided bounds, and various experimental clarifications for the results. The authors did a strong effort to address comments. Unfortunately, several concerns remain that, even after rebuttal, the reviewer who participated in the discussion did not seem convinced (but raised the score), and hence, it's very uncertain if in the end the reviewer would raise even more the scores. The reviewer who provided a very strong score, did not participate in discussion and review is a bit weak. Overall, even if another reviewer would raise their scores, the paper's overall score would be a borderline case and very uncertain it would finally be accepted. I hope the authors would address the comments and find a new venue for the work.

**Reviewer Concerns:**

Outstanding concerns remain about the novelty and inclusion of baselines. it's unclear if the reviewers would have been convinced after the discussion.

**Reviewer Scores:**

Even after assuming a raise of scores by half the reviewers, the overall score would be in borderline case and hence uncertain it would be accepted in the end.

---

### Decision · Program_Chairs · 2026-01-26

Reject